# On the Global Linear Convergence of Frank-Wolfe Optimization Variants

**Simon Lacoste-Julien**
INRIA - SIERRA project-team
École Normale Supérieure, Paris, France

**Martin Jaggi**
Dept. of Computer Science
ETH Zürich, Switzerland

## Abstract

The Frank-Wolfe (FW) optimization algorithm has lately re-gained popularity thanks in particular to its ability to nicely handle the structured constraints appearing in machine learning applications. However, its convergence rate is known to be slow (sublinear) when the solution lies at the boundary. A simple less-known fix is to add the possibility to take 'away steps' during optimization, an operation that importantly *does not* require a feasibility oracle. In this paper, we highlight and clarify several variants of the Frank-Wolfe optimization algorithm that have been successfully applied in practice: away-steps FW, pairwise FW, fully-corrective FW and Wolfe's minimum norm point algorithm, and prove for the first time that they all enjoy global linear convergence, under a weaker condition than strong convexity of the objective. The constant in the convergence rate has an elegant interpretation as the product of the (classical) condition number of the function with a novel geometric quantity that plays the role of a 'condition number' of the constraint set. We provide pointers to where these algorithms have made a difference in practice, in particular with the flow polytope, the marginal polytope and the base polytope for submodular optimization.

The Frank-Wolfe algorithm [9] (also known as *conditional gradient*) is one of the earliest existing methods for constrained convex optimization, and has seen an impressive revival recently due to its nice properties compared to projected or proximal gradient methods, in particular for sparse optimization and machine learning applications.

On the other hand, the classical projected gradient and proximal methods have been known to exhibit a very nice adaptive acceleration property, namely that the the convergence rate becomes linear for strongly convex objective, i.e. that the optimization error of the same algorithm after $t$ iterations will decrease geometrically with $O((1 - \rho)^t)$ instead of the usual $O(1/t)$ for general convex objective functions. It has become an active research topic recently whether such an acceleration is also possible for Frank-Wolfe type methods.

**Contributions.** We clarify several variants of the Frank-Wolfe algorithm and show that they all converge linearly for any strongly convex function optimized over a polytope domain, with a constant bounded away from zero that only depends on the geometry of the polytope. Our analysis does *not* depend on the location of the true optimum with respect to the domain, which was a disadvantage of earlier existing results such as [34, 12, 5], and the newer work of [28], as well as the line of work of [1, 19, 26] which rely on Robinson's condition [30]. Our analysis yields a weaker sufficient condition than Robinson's condition; in particular we can have linear convergence even in some cases when the function has more than one global minima, and is not globally strongly convex. The constant also naturally separates as the product of the condition number of the function with a novel notion of condition number of a polytope, which might have applications in complexity theory.

**Related Work.** For the classical Frank-Wolfe algorithm, [5] showed a linear rate for the special case of quadratic objectives when the optimum is in the strict interior of the domain, a result already subsumed by the more general [12]. The early work of [23] showed linear convergence for *strongly*

*convex constraint sets*, under the strong requirement that the gradient norm is not too small (see [11] for a discussion). The away-steps variant of the Frank-Wolfe algorithm, that can also remove weight from 'bad' atoms in the current active set, was proposed in [34], and later also analyzed in [12]. The precise method is stated below in Algorithm 1. [12] showed a (local) linear convergence rate on polytopes, but the constant unfortunately depends on the distance between the solution and its relative boundary, a quantity that can be arbitrarily small. More recently, [1, 19, 26] have obtained linear convergence results in the case that the optimum solution satisfies Robinson's condition [30]. In a different recent line of work, [10, 22] have studied a variation of FW that repeatedly moves mass from the worst vertices to the standard FW vertex until a specific condition is satisfied, yielding a linear rate on strongly convex functions. Their algorithm requires the knowledge of several constants though, and moreover is not adaptive to the best-case scenario, unlike the Frank-Wolfe algorithm with away steps and line-search. None of these previous works was shown to be affine invariant, and most require additional knowledge about problem specific parameters.

**Setup.**    We consider general constrained convex optimization problems of the form:

$$\min_{\boldsymbol{x} \in \mathcal{M}} f(\boldsymbol{x}), \qquad \mathcal{M} = \text{conv}(\mathcal{A}), \qquad \text{with only access to:} \quad \text{LMO}_{\mathcal{A}}(\boldsymbol{r}) \in \arg\min_{\boldsymbol{x} \in \mathcal{A}} \langle \boldsymbol{r}, \boldsymbol{x} \rangle, \quad (1)$$

where $\mathcal{A} \subseteq \mathbb{R}^d$ is a *finite* set of vectors that we call *atoms*.[1] We assume that the function $f$ is $\mu$-strongly convex with $L$-Lipschitz continuous gradient over $\mathcal{M}$. We also consider weaker conditions than strong convexity for $f$ in Section 4. As $\mathcal{A}$ is finite, $\mathcal{M}$ is a (convex and bounded) polytope. The methods that we consider in this paper only require access to a *linear minimization oracle* $\text{LMO}_{\mathcal{A}}(.)$ associated with the domain $\mathcal{M}$ through a generating set of atoms $\mathcal{A}$. This oracle is defined as to return a minimizer of a linear subproblem over $\mathcal{M} = \text{conv}(\mathcal{A})$, for any given direction $\boldsymbol{r} \in \mathbb{R}^d$.[2]

**Examples.**    Optimization problems of the form (1) appear widely in machine learning and signal processing applications. The set of atoms $\mathcal{A}$ can represent combinatorial objects of arbitrary type. Efficient linear minimization oracles often exist in the form of dynamic programs or other combinatorial optimization approaches. As an example from tracking in computer vision, $\mathcal{A}$ could be the set of integer flows on a graph [16, 7], where $\text{LMO}_{\mathcal{A}}$ can be efficiently implemented by a minimum cost network flow algorithm. In this case, $\mathcal{M}$ can also be described with a polynomial number of linear inequalities. But in other examples, $\mathcal{M}$ might not have a polynomial description in terms of linear inequalities, and testing membership in $\mathcal{M}$ might be much more expensive than running the linear oracle. This is the case when optimizing over the *base polytope*, an object appearing in submodular function optimization [3]. There, the $\text{LMO}_{\mathcal{A}}$ oracle is a simple greedy algorithm. Another example is when $\mathcal{A}$ represents the possible consistent value assignments on cliques of a Markov random field (MRF); $\mathcal{M}$ is the *marginal polytope* [32], where testing membership is NP-hard in general, though efficient linear oracles exist for some special cases [17]. Optimization over the marginal polytope appears for example in structured SVM learning [21] and variational inference [18].

**The Original Frank-Wolfe Algorithm.**    The Frank-Wolfe (FW) optimization algorithm [9], also known as *conditional gradient* [23], is particularly suited for the setup (1) where $\mathcal{M}$ is only accessed through the linear minimization oracle. It works as follows: At a current iterate $\boldsymbol{x}^{(t)}$, the algorithm finds a feasible search atom $\boldsymbol{s}_t$ to move towards by minimizing the linearization of the objective function $f$ over $\mathcal{M}$ (line 3 in Algorithm 1) – this is where the linear minimization oracle $\text{LMO}_{\mathcal{A}}$ is used. The next iterate $\boldsymbol{x}^{(t+1)}$ is then obtained by doing a line-search on $f$ between $\boldsymbol{x}^{(t)}$ and $\boldsymbol{s}_t$ (line 11 in Algorithm 1). One reason for the recent increased popularity of Frank-Wolfe-type algorithms is the sparsity of their iterates: in iteration $t$ of the algorithm, the iterate can be represented as a sparse convex combination of at most $t + 1$ atoms $\mathcal{S}^{(t)} \subseteq \mathcal{A}$ of the domain $\mathcal{M}$, which we write as $\boldsymbol{x}^{(t)} = \sum_{\boldsymbol{v} \in \mathcal{S}^{(t)}} \alpha_{\boldsymbol{v}}^{(t)} \boldsymbol{v}$. We write $\mathcal{S}^{(t)}$ for the *active set*, containing the previously discovered search atoms $\boldsymbol{s}_r$ for $r < t$ that have non-zero *weight* $\alpha_{\boldsymbol{s}_r}^{(t)} > 0$ in the expansion (potentially also including the starting point $\boldsymbol{x}^{(0)}$). While tracking the active set $\mathcal{S}^{(t)}$ is not necessary for the original FW algorithm, the improved variants of FW that we discuss will require that $\mathcal{S}^{(t)}$ is maintained.

**Zig-Zagging Phenomenon.**    When the optimal solution lies at the boundary of $\mathcal{M}$, the convergence rate of the iterates is slow, i.e. sublinear: $f(\boldsymbol{x}^{(t)}) - f(\boldsymbol{x}^*) \leq O(1/t)$, for $\boldsymbol{x}^*$ being an optimal solution [9, 6, 8, 15]. This is because the iterates of the classical FW algorithm start to zig-zag

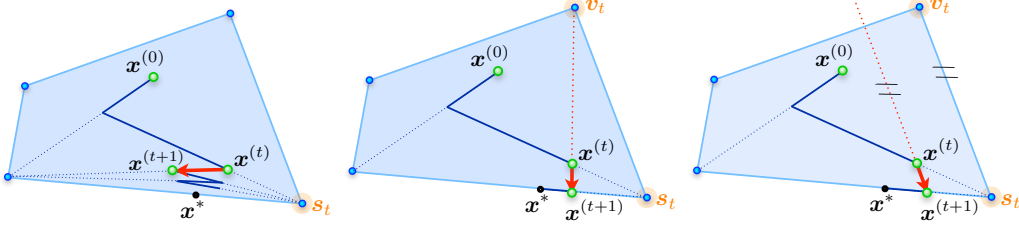

Figure 1: (left) The FW algorithm zig-zags when the solution $x^*$ lies on the boundary. (middle) Adding the possibility of an *away step* attenuates this problem. (right) As an alternative, a pairwise FW step.

between the vertices defining the face containing the solution $x^*$ (see left of Figure 1). In fact, the $1/t$ rate is tight for a large class of functions: Canon and Cullum [6], Wolfe [34] showed (roughly) that $f(x^{(t)}) - f(x^*) \geq \Omega(1/t^{1+\delta})$ for any $\delta > 0$ when $x^*$ lies on a face of $\mathcal{M}$ with some additional regularity assumptions. Note that this lower bound is different than the $\Omega(1/t)$ one presented in [15, Lemma 3] which holds for all one-atom-per-step algorithms but assumes high dimensionality $d \geq t$.

# 1 Improved Variants of the Frank-Wolfe Algorithm

---

**Algorithm 1** Away-steps Frank-Wolfe algorithm: $\mathbf{AFW}(x^{(0)}, \mathcal{A}, \epsilon)$

---

1: Let $x^{(0)} \in \mathcal{A}$, and $\mathcal{S}^{(0)} := \{x^{(0)}\}$      *(so that $\alpha_v^{(0)} = 1$ for $v = x^{(0)}$ and 0 otherwise)*
2: **for** $t = 0 \ldots T$ **do**
3:      Let $s_t := \mathrm{LMO}_{\mathcal{A}}(\nabla f(x^{(t)}))$ and $d_t^{\mathrm{FW}} := s_t - x^{(t)}$      *(the FW direction)*
4:      Let $v_t \in \arg\max_{v \in \mathcal{S}^{(t)}} \langle \nabla f(x^{(t)}), v \rangle$ and $d_t^{\mathrm{A}} := x^{(t)} - v_t$      *(the away direction)*
5:      **if** $g_t^{\mathrm{FW}} := \langle -\nabla f(x^{(t)}), d_t^{\mathrm{FW}} \rangle \leq \epsilon$ **then return** $x^{(t)}$      *(FW gap is small enough, so return)*
6:      **if** $\langle -\nabla f(x^{(t)}), d_t^{\mathrm{FW}} \rangle \geq \langle -\nabla f(x^{(t)}), d_t^{\mathrm{A}} \rangle$ **then**
7:          $d_t := d_t^{\mathrm{FW}}$, and $\gamma_{\max} := 1$      *(choose the FW direction)*
8:      **else**
9:          $d_t := d_t^{\mathrm{A}}$, and $\gamma_{\max} := \alpha_{v_t}/(1 - \alpha_{v_t})$      *(choose away direction; maximum feasible step-size)*
10:      **end if**
11:      Line-search: $\gamma_t \in \arg\min_{\gamma \in [0, \gamma_{\max}]} f(x^{(t)} + \gamma d_t)$
12:      Update $x^{(t+1)} := x^{(t)} + \gamma_t d_t$      *(and accordingly for the weights $\alpha^{(t+1)}$, see text)*
13:      Update $\mathcal{S}^{(t+1)} := \{v \in \mathcal{A} \text{ s.t. } \alpha_v^{(t+1)} > 0\}$
14: **end for**

---

**Algorithm 2** Pairwise Frank-Wolfe algorithm: $\mathbf{PFW}(x^{(0)}, \mathcal{A}, \epsilon)$

---

1: ... as in Algorithm 1, except replacing lines 6 to 10 by: $d_t = d_t^{\mathrm{PFW}} := s_t - v_t$, and $\gamma_{\max} := \alpha_{v_t}$.

---

**Away-Steps Frank-Wolfe.** To address the zig-zagging problem of FW, Wolfe [34] proposed to add the possibility to move *away* from an active atom in $\mathcal{S}^{(t)}$ (see middle of Figure 1); this simple modification is sufficient to make the algorithm linearly convergent for strongly convex functions. We describe the away-steps variant of Frank-Wolfe in Algorithm 1.[3] The *away* direction $d_t^{\mathrm{A}}$ is defined in line 4 by finding the atom $v_t$ in $\mathcal{S}^{(t)}$ that maximizes the potential of descent given by $g_t^{\mathrm{A}} := \langle -\nabla f(x^{(t)}), x^{(t)} - v_t \rangle$. Note that this search is over the (typically small) active set $\mathcal{S}^{(t)}$, and is fundamentally easier than the linear oracle $\mathrm{LMO}_{\mathcal{A}}$. The maximum step-size $\gamma_{\max}$ as defined on line 9 ensures that the new iterate $x^{(t)} + \gamma d_t^{\mathrm{A}}$ stays in $\mathcal{M}$. In fact, this guarantees that the convex representation is maintained, and we stay inside $\mathrm{conv}(\mathcal{S}^{(t)}) \subseteq \mathcal{M}$. When $\mathcal{M}$ is a simplex, then the barycentric coordinates are unique and $x^{(t)} + \gamma_{\max} d_t^{\mathrm{A}}$ truly lies on the boundary of $\mathcal{M}$. On the other hand, if $|\mathcal{A}| > \dim(\mathcal{M}) + 1$ (e.g. for the cube), then it could hypothetically be possible to have a step-size bigger than $\gamma_{\max}$ which is still feasible. Computing the true maximum feasible step-size would require the ability to know when we cross the boundary of $\mathcal{M}$ along a specific line, which is not possible for general $\mathcal{M}$. Using the conservative maximum step-size of line 9 ensures that we

do not need this more powerful oracle. This is why Algorithm 1 requires to maintain $\mathcal{S}^{(t)}$ (unlike standard FW). Finally, as in classical FW, the FW gap $g_t^{\text{FW}}$ is an upper bound on the unknown suboptimality, and can be used as a stopping criterion:

$$g_t^{\text{FW}} := \left\langle -\nabla f(\boldsymbol{x}^{(t)}), \boldsymbol{d}_t^{\text{FW}} \right\rangle \geq \left\langle -\nabla f(\boldsymbol{x}^{(t)}), \boldsymbol{x}^* - \boldsymbol{x}^{(t)} \right\rangle \geq f(\boldsymbol{x}^{(t)}) - f(\boldsymbol{x}^*) \quad \text{(by convexity)}.$$

If $\gamma_t = \gamma_{\max}$, then we call this step a *drop step*, as it fully removes the atom $\boldsymbol{v}_t$ from the currently active set of atoms $\mathcal{S}^{(t)}$ (by settings its weight to zero). The weight updates for lines 12 and 13 are of the following form: For a FW step, we have $\mathcal{S}^{(t+1)} = \{\boldsymbol{s}_t\}$ if $\gamma_t = 1$; otherwise $\mathcal{S}^{(t+1)} = \mathcal{S}^{(t)} \cup \{\boldsymbol{s}_t\}$. Also, we have $\alpha_{\boldsymbol{s}_t}^{(t+1)} := (1-\gamma_t)\alpha_{\boldsymbol{s}_t}^{(t)} + \gamma_t$ and $\alpha_{\boldsymbol{v}}^{(t+1)} := (1-\gamma_t)\alpha_{\boldsymbol{v}}^{(t)}$ for $\boldsymbol{v} \in \mathcal{S}^{(t)} \setminus \{\boldsymbol{s}_t\}$. For an away step, we have $\mathcal{S}^{(t+1)} = \mathcal{S}^{(t)} \setminus \{\boldsymbol{v}_t\}$ if $\gamma_t = \gamma_{\max}$ (a *drop step*); otherwise $\mathcal{S}^{(t+1)} = \mathcal{S}^{(t)}$. Also, we have $\alpha_{\boldsymbol{v}_t}^{(t+1)} := (1 + \gamma_t)\alpha_{\boldsymbol{v}_t}^{(t)} - \gamma_t$ and $\alpha_{\boldsymbol{v}}^{(t+1)} := (1 + \gamma_t)\alpha_{\boldsymbol{v}}^{(t)}$ for $\boldsymbol{v} \in \mathcal{S}^{(t)} \setminus \{\boldsymbol{v}_t\}$.

**Pairwise Frank-Wolfe.** The next variant that we present is inspired by an early algorithm by Mitchell et al. [25], called the MDM algorithm, originally invented for the polytope distance problem. Here the idea is to only move weight mass between two atoms in each step. More precisely, the generalized method as presented in Algorithm 2 moves weight from the away atom $\boldsymbol{v}_t$ to the FW atom $\boldsymbol{s}_t$, and keeps all other $\alpha$ weights un-changed. We call such a swap of mass between the two atoms a *pairwise FW* step, i.e. $\alpha_{\boldsymbol{v}_t}^{(t+1)} = \alpha_{\boldsymbol{v}_t}^{(t)} - \gamma$ and $\alpha_{\boldsymbol{s}_t}^{(t+1)} = \alpha_{\boldsymbol{s}_t}^{(t)} + \gamma$ for some step-size $\gamma \leq \gamma_{\max} := \alpha_{\boldsymbol{v}_t}^{(t)}$. In contrast, classical FW shrinks all active weights at every iteration.

The pairwise FW direction will also be central to our proof technique to provide the first global linear convergence rate for away-steps FW, as well as the fully-corrective variant and Wolfe's min-norm-point algorithm.

As we will see in Section 2.2, the rate guarantee for the pairwise FW variant is more loose than for the other variants, because we cannot provide a satisfactory bound on the number of the problematic *swap steps* (defined just before Theorem 1). Nevertheless, the algorithm seems to perform quite well in practice, often outperforming away-steps FW, especially in the important case of sparse solutions, that is if the optimal solution $\boldsymbol{x}^*$ lies on a low-dimensional face of $\mathcal{M}$ (and thus one wants to keep the active set $\mathcal{S}^{(t)}$ small). The pairwise FW step is arguably more efficient at pruning the coordinates in $\mathcal{S}^{(t)}$. In contrast to the away step which moves the mass back *uniformly* onto all other active elements $\mathcal{S}^{(t)}$ (and might require more corrections later), the pairwise FW step only moves the mass onto the (good) FW atom $\boldsymbol{s}_t$. A slightly different version than Algorithm 2 was also proposed by Ñanculef et al. [26], though their convergence proofs were incomplete (see Appendix A.3). The algorithm is related to classical working set algorithms, such as the SMO algorithm used to train SVMs [29]. We refer to [26] for an empirical comparison for SVMs, as well as their Section 5 for more related work. See also Appendix A.3 for a link between pairwise FW and [10].

**Fully-Corrective Frank-Wolfe, and Wolfe's Min-Norm Point Algorithm.** When the linear oracle is expensive, it might be worthwhile to do more work to optimize over the active set $\mathcal{S}^{(t)}$ in between each call to the linear oracle, rather than just performing an away or pairwise step. We give in Algorithm 3 the fully-corrective Frank-Wolfe (FCFW) variant, that maintains a correction polytope defined by a set of atoms $\mathcal{A}^{(t)}$ (potentially larger than the active set $\mathcal{S}^{(t)}$). Rather than obtaining the next iterate by line-search, $\boldsymbol{x}^{(t+1)}$ is obtained by re-optimizing $f$ over $\text{conv}(\mathcal{A}^{(t)})$. Depending on how the correction is implemented, and how the correction atoms $\mathcal{A}^{(t)}$ are maintained, several variants can be obtained. These variants are known under many names, such as the extended FW method by Holloway [14] or the simplicial decomposition method [31, 13]. Wolfe's min-norm point (MNP) algorithm [35] for polytope distance problems is often confused with FCFW for quadratic objectives. The major difference is that standard FCFW optimizes $f$ over $\text{conv}(\mathcal{A}^{(t)})$, whereas MNP implements the correction as a sequence of affine projections that potentially yield a different update, but can be computed more efficiently in several practical applications [35]. We describe precisely in Appendix A.1 a generalization of the MNP algorithm as a specific case of the correction subroutine from step 7 of the generic Algorithm 3.

The original convergence analysis of the FCFW algorithm [14] (and also MNP algorithm [35]) only showed that they were finitely convergent, with a bound on the number of iterations in terms of the cardinality of $\mathcal{A}$ (unfortunately an exponential number in general). Holloway [14] also argued that FCFW had an asymptotic linear convergence based on the flawed argument of Wolfe [34]. As far as we know, our work is the first to provide global linear convergence rates for FCFW and MNP for

---

**Algorithm 3** Fully-corrective Frank-Wolfe with approximate correction: $\mathbf{FCFW}(\boldsymbol{x}^{(0)}, \mathcal{A}, \epsilon)$

---

1: **Input:** Set of atoms $\mathcal{A}$, active set $\mathcal{S}^{(0)}$, starting point $\boldsymbol{x}^{(0)} = \sum_{\boldsymbol{v} \in \mathcal{S}^{(0)}} \alpha_{\boldsymbol{v}}^{(0)} \boldsymbol{v}$, stopping criterion $\epsilon$.

2: Let $\mathcal{A}^{(0)} := \mathcal{S}^{(0)}$    (optionally, a bigger $\mathcal{A}^{(0)}$ could be passed as argument for a warm start)
3: **for** $t = 0 \dots T$ **do**
4:    Let $\boldsymbol{s}_t := \mathrm{LMO}_{\mathcal{A}}\big(\nabla f(\boldsymbol{x}^{(t)})\big)$              *(the FW atom)*
5:    Let $\boldsymbol{d}_t^{\mathrm{FW}} := \boldsymbol{s}_t - \boldsymbol{x}^{(t)}$ and $g_t^{\mathrm{FW}} = \big\langle -\nabla f(\boldsymbol{x}^{(t)}), \boldsymbol{d}_t^{\mathrm{FW}} \big\rangle$       *(FW gap)*
6:    **if** $g_t^{\mathrm{FW}} \le \epsilon$ **then return** $\boldsymbol{x}^{(t)}$
7:    $(\boldsymbol{x}^{(t+1)}, \mathcal{A}^{(t+1)}) := \mathbf{Correction}(\boldsymbol{x}^{(t)}, \mathcal{A}^{(t)}, \boldsymbol{s}_t, \epsilon)$       *(approximate correction step)*
8: **end for**

---

**Algorithm 4** Approximate correction: $\mathbf{Correction}(\boldsymbol{x}^{(t)}, \mathcal{A}^{(t)}, \boldsymbol{s}_t, \epsilon)$

---

1: Return $(\boldsymbol{x}^{(t+1)}, \mathcal{A}^{(t+1)})$ with the following properties:
2:    $\mathcal{S}^{(t+1)}$ is the active set for $\boldsymbol{x}^{(t+1)}$ and $\mathcal{A}^{(t+1)} \supseteq \mathcal{S}^{(t+1)}$.
3:    $f(\boldsymbol{x}^{(t+1)}) \le \min_{\gamma \in [0,1]} f\big(\boldsymbol{x}^{(t)} + \gamma(\boldsymbol{s}_t - \boldsymbol{x}^{(t)})\big)$       *(make at least as much progress as a FW step)*
4:    $g_{t+1}^{\mathrm{A}} := \max_{\boldsymbol{v} \in \mathcal{S}^{(t+1)}} \big\langle -\nabla f(\boldsymbol{x}^{(t+1)}), \boldsymbol{x}^{(t+1)} - \boldsymbol{v} \big\rangle \le \epsilon$    *(the away gap is small enough)*

---

general strongly convex functions. Moreover, the proof of convergence for FCFW does not require an exact solution to the correction step; instead, we show that the weaker properties stated for the approximate correction procedure in Algorithm 4 are sufficient for a global linear convergence rate (this correction could be implemented using away-steps FW, as done for example in [18]).

## 2    Global Linear Convergence Analysis

### 2.1    Intuition for the Convergence Proofs

We first give the general intuition for the linear convergence proof of the different FW variants, starting from the work of Guélat and Marcotte [12]. We assume that the objective function $f$ is smooth over a compact set $\mathcal{M}$, i.e. its gradient is Lipschitz continuous with constant $L$. Also let $M := \mathrm{diam}(\mathcal{M})$. Let $\boldsymbol{d}_t$ be the direction in which the line-search is executed by the algorithm (Line 11 in Algorithm 1). By the standard descent lemma [see e.g. (1.2.5) in 27], we have:

$$f(\boldsymbol{x}^{(t+1)}) \le f(\boldsymbol{x}^{(t)} + \gamma \boldsymbol{d}_t) \le f(\boldsymbol{x}^{(t)}) + \gamma \left\langle \nabla f(\boldsymbol{x}^{(t)}), \boldsymbol{d}_t \right\rangle + \frac{\gamma^2}{2} L \|\boldsymbol{d}_t\|^2 \quad \forall \gamma \in [0, \gamma_{\max}]. \quad (2)$$

We let $\boldsymbol{r}_t := -\nabla f(\boldsymbol{x}^{(t)})$ and let $h_t := f(\boldsymbol{x}^{(t)}) - f(\boldsymbol{x}^*)$ be the suboptimality error. Supposing for now that $\gamma_{\max} \ge \gamma_t^* := \langle \boldsymbol{r}_t, \boldsymbol{d}_t \rangle / (L \|\boldsymbol{d}_t\|^2)$. We can set $\gamma = \gamma_t^*$ to minimize the RHS of (2), subtract $f(\boldsymbol{x}^*)$ on both sides, and re-organize to get a lower bound on the progress:

$$h_t - h_{t+1} \ge \frac{\langle \boldsymbol{r}_t, \boldsymbol{d}_t \rangle^2}{2L \|\boldsymbol{d}_t\|^2} = \frac{1}{2L} \langle \boldsymbol{r}_t, \hat{\boldsymbol{d}}_t \rangle^2, \quad (3)$$

where we use the 'hat' notation to denote normalized vectors: $\hat{\boldsymbol{d}}_t := \boldsymbol{d}_t / \|\boldsymbol{d}_t\|$. Let $\boldsymbol{e}_t := \boldsymbol{x}^* - \boldsymbol{x}^{(t)}$ be the error vector. By $\mu$-strong convexity of $f$, we have:

$$f(\boldsymbol{x}^{(t)} + \gamma \boldsymbol{e}_t) \ge f(\boldsymbol{x}^{(t)}) + \gamma \left\langle \nabla f(\boldsymbol{x}^{(t)}), \boldsymbol{e}_t \right\rangle + \frac{\gamma^2}{2} \mu \|\boldsymbol{e}_t\|^2 \quad \forall \gamma \in [0, 1]. \quad (4)$$

The RHS is lower bounded by its minimum as a function of $\gamma$ (unconstrained), achieved using $\gamma := \langle \boldsymbol{r}_t, \boldsymbol{e}_t \rangle / (\mu \|\boldsymbol{e}_t\|^2)$. We are then free to use any value of $\gamma$ on the LHS and maintain a valid bound. In particular, we use $\gamma = 1$ to obtain $f(\boldsymbol{x}^*)$. Again re-arranging, we get:

$$h_t \le \frac{\langle \boldsymbol{r}_t, \hat{\boldsymbol{e}}_t \rangle^2}{2\mu}, \quad \text{and combining with (3), we obtain:} \quad h_t - h_{t+1} \ge \frac{\mu}{L} \frac{\langle \boldsymbol{r}_t, \hat{\boldsymbol{d}}_t \rangle^2}{\langle \boldsymbol{r}_t, \hat{\boldsymbol{e}}_t \rangle^2} h_t. \quad (5)$$

The inequality (5) is fairly general and valid for any line-search method in direction $\boldsymbol{d}_t$. To get a linear convergence rate, we need to lower bound (by a positive constant) the term in front of $h_t$ on the RHS, which depends on the angle between the update direction $\boldsymbol{d}_t$ and the negative gradient $\boldsymbol{r}_t$. If we assume that the solution $\boldsymbol{x}^*$ lies in the relative interior of $\mathcal{M}$ with a distance of at least $\delta > 0$ from the boundary, then $\langle \boldsymbol{r}_t, \boldsymbol{d}_t \rangle \ge \delta \|\boldsymbol{r}_t\|$ for the FW direction $\boldsymbol{d}_t^{\mathrm{FW}}$, and by combining with $\|\boldsymbol{d}_t\| \le M$, we get a linear rate with constant $1 - \frac{\mu}{L}(\frac{\delta}{M})^2$ (this was the result from [12]). On the other hand, if $\boldsymbol{x}^*$ lies on the boundary, then $\langle \hat{\boldsymbol{r}}_t, \hat{\boldsymbol{d}}_t \rangle$ gets arbitrary close to zero for standard FW (the zig-zagging phenomenon) and the convergence is sublinear.

**Proof Sketch for AFW.** The key insight to prove the global linear convergence for AFW is to relate $\langle r_t, d_t \rangle$ with the *pairwise FW* direction $d_t^{\text{PFW}} := s_t - v_t$. By the way the direction $d_t$ is chosen on lines 6 to 10 of Algorithm 1, we have:

$$2\langle r_t, d_t \rangle \geq \langle r_t, d_t^{\text{FW}} \rangle + \langle r_t, d_t^A \rangle = \langle r_t, d_t^{\text{FW}} + d_t^A \rangle = \langle r_t, d_t^{\text{PFW}} \rangle. \tag{6}$$

We thus have $\langle r_t, d_t \rangle \geq \langle r_t, d_t^{\text{PFW}} \rangle / 2$. Now the crucial property of the pairwise FW direction is that for any potential negative gradient direction $r_t$, the worst case inner product $\langle \hat{r}_t, d_t^{\text{PFW}} \rangle$ can be lower bounded away from zero by a quantity depending only on the geometry of $\mathcal{M}$ (unless we are at the optimum). We call this quantity the *pyramidal width* of $\mathcal{A}$. The figure on the right shows the six possible pairwise FW directions $d_t^{\text{PFW}}$ for a triangle domain, depending on which colored area the $r_t$ direction falls into. We will see that the pyramidal width is related to the smallest width of pyramids that we can construct from $\mathcal{A}$ in a specific way related to the choice of the away and towards atoms $v_t$ and $s_t$. See (9) and our main Theorem 3 in Section 3.

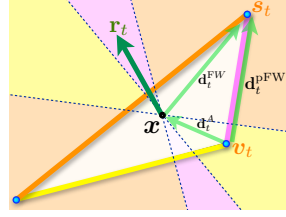

This gives the main argument for the linear convergence of AFW for steps where $\gamma_t^* \leq \gamma_{\max}$. When $\gamma_{\max}$ is too small, AFW will perform a *drop step*, as the line-search will truncate the step-size to $\gamma_t = \gamma_{\max}$. We cannot guarantee sufficient progress in this case, but the drop step decreases the active set size by one, and thus they cannot happen too often (not more than half the time). These are the main elements for the global linear convergence proof for AFW. The rest is to carefully consider various boundary cases. We can re-use the same techniques to prove the convergence for pairwise FW, though unfortunately the latter also has the possibility of problematic *swap steps*. While their number can be bounded, so far we only found the extremely loose bound quoted in Theorem 1.

**Proof Sketch for FCFW.** For FCFW, by line 4 of the correction Algorithm 4, the away gap satisfies $g_t^A \leq \epsilon$ at the beginning of a new iteration. Supposing that the algorithm does not exit at line 6 of Algorithm 3, we have $g_t^{\text{FW}} > \epsilon$ and therefore $2\langle r_t, d_t^{\text{FW}} \rangle \geq \langle r_t, d_t^{\text{PFW}} \rangle$ using a similar argument as in (6). Finally, by line 3 of Algorithm 4, the correction is guaranteed to make at least as much progress as a line-search in direction $d_t^{\text{FW}}$, and so the progress bound (5) applies also to FCFW.

## 2.2 Convergence Results

We now give the global linear convergence rates for the four variants of the FW algorithm: away-steps FW (AFW Alg. 1); pairwise FW (PFW Alg. 2); fully-corrective FW (FCFW Alg. 3 with approximate correction Alg. 4); and Wolfe's min-norm point algorithm (Alg. 3 with MNP-correction as Alg. 5 in Appendix A.1). For the AFW, MNP and PFW algorithms, we call a *drop step* when the active set shrinks $|S^{(t+1)}| < |S^{(t)}|$. For the PFW algorithm, we also have the possibility of a *swap step* where $\gamma_t = \gamma_{\max}$ but $|S^{(t+1)}| = |S^{(t)}|$ (i.e. the mass was fully swapped from the away atom to the FW atom). A nice property of FCFW is that it does not have any drop step (it executes both FW steps and away steps simultaneously while guaranteeing enough progress at every iteration).

**Theorem 1.** *Suppose that $f$ has $L$-Lipschitz gradient[4] and is $\mu$-strongly convex over $\mathcal{M} = \operatorname{conv}(\mathcal{A})$. Let $M = \operatorname{diam}(\mathcal{M})$ and $\delta = PWidth(\mathcal{A})$ as defined by (9). Then the suboptimality $h_t$ of the iterates of all the four variants of the FW algorithm decreases geometrically at each step that is not a drop step nor a swap step (i.e. when $\gamma_t < \gamma_{\max}$, called a 'good step'), that is*

$$h_{t+1} \leq (1 - \rho) h_t, \qquad \text{where } \rho := \frac{\mu}{4L} \left( \frac{\delta}{M} \right)^2.$$

*Let $k(t)$ be the number of 'good steps' up to iteration $t$. We have $k(t) = t$ for FCFW; $k(t) \geq t/2$ for MNP and AFW; and $k(t) \geq t/(3|\mathcal{A}|! + 1)$ for PFW (because of the swap steps). This yields a global linear convergence rate of $h_t \leq h_0 \exp(-\rho \, k(t))$ for all variants. If $\mu = 0$ (general convex), then $h_t = O(1/k(t))$ instead. See Theorem 8 in Appendix D for an affine invariant version and proof.*

Note that to our knowledge, none of the existing linear convergence results showed that the duality gap was also linearly convergent. The result for the gap follows directly from the simple manipulation of (2); putting the FW gap to the LHS and optimizing the RHS for $\gamma \in [0, 1]$.

**Theorem 2.** *Suppose that $f$ has $L$-Lipschitz gradient over $\mathcal{M}$ with $M := \operatorname{diam}(\mathcal{M})$. Then the FW gap $g_t^{\text{FW}}$ for any algorithm is upper bounded by the primal error $h_t$ as follows:*

$$g_t^{\text{FW}} \leq h_t + LM^2/2 \text{ when } h_t > LM^2/2, \qquad g_t^{\text{FW}} \leq M\sqrt{2h_t L} \text{ otherwise}. \tag{7}$$

# 3 Pyramidal Width

We now describe the claimed lower bound on the angle between the negative gradient and the pairwise FW direction, which depends only on the geometric properties of $\mathcal{M}$. According to our argument about the progress bound (5) and the PFW gap (6), our goal is to find a lower bound on $\langle \boldsymbol{r}_t, \boldsymbol{d}_t^{\text{PFW}} \rangle / \langle \boldsymbol{r}_t, \hat{\boldsymbol{e}}_t \rangle$. First note that $\langle \boldsymbol{r}_t, \boldsymbol{d}_t^{\text{PFW}} \rangle = \langle \boldsymbol{r}_t, \boldsymbol{s}_t - \boldsymbol{v}_t \rangle = \max_{\boldsymbol{s} \in \mathcal{M}, \boldsymbol{v} \in \mathcal{S}^{(t)}} \langle \boldsymbol{r}_t, \boldsymbol{s} - \boldsymbol{v} \rangle$ where $\mathcal{S}^{(t)}$ is a possible active set for $\boldsymbol{x}^{(t)}$. This looks like the *directional width* of a pyramid with base $\mathcal{S}^{(t)}$ and summit $\boldsymbol{s}_t$. To be conservative, we consider the worst case possible active set for $\boldsymbol{x}^{(t)}$; this is what we will call the *pyramid directional width* $PdirW(\mathcal{A}, \boldsymbol{r}_t, \boldsymbol{x}^{(t)})$. We start with the following definitions.

**Directional Width.** The directional width of a set $\mathcal{A}$ with respect to a direction $\boldsymbol{r}$ is defined as $dirW(\mathcal{A}, \boldsymbol{r}) := \max_{\boldsymbol{s}, \boldsymbol{v} \in \mathcal{A}} \left\langle \frac{\boldsymbol{r}}{\|\boldsymbol{r}\|}, \boldsymbol{s} - \boldsymbol{v} \right\rangle$. The *width* of $\mathcal{A}$ is the minimum directional width over all possible directions in its affine hull.

**Pyramidal Directional Width.** We define the pyramidal directional width of a set $\mathcal{A}$ with respect to a direction $\boldsymbol{r}$ and a base point $\boldsymbol{x} \in \mathcal{M}$ to be

$$PdirW(\mathcal{A}, \boldsymbol{r}, \boldsymbol{x}) := \min_{\mathcal{S} \in \mathcal{S}_{\boldsymbol{x}}} dirW(\mathcal{S} \cup \{\boldsymbol{s}(\mathcal{A}, \boldsymbol{r})\}, \ \boldsymbol{r}) = \min_{\mathcal{S} \in \mathcal{S}_{\boldsymbol{x}}} \max_{\boldsymbol{s} \in \mathcal{A}, \boldsymbol{v} \in \mathcal{S}} \left\langle \frac{\boldsymbol{r}}{\|\boldsymbol{r}\|}, \boldsymbol{s} - \boldsymbol{v} \right\rangle, \quad (8)$$

where $\mathcal{S}_{\boldsymbol{x}} := \{\mathcal{S} \,|\, \mathcal{S} \subseteq \mathcal{A}$ such that $\boldsymbol{x}$ is a proper[5] convex combination of all the elements in $\mathcal{S}\}$, and $\boldsymbol{s}(\mathcal{A}, \boldsymbol{r}) := \arg\max_{\boldsymbol{v} \in \mathcal{A}} \langle \boldsymbol{r}, \boldsymbol{v} \rangle$ is the FW atom used as a summit.

**Pyramidal Width.** To define the pyramidal width of a set, we take the minimum over the cone of possible *feasible* directions $\boldsymbol{r}$ (in order to avoid the problem of zero width).
A direction $\boldsymbol{r}$ is *feasible* for $\mathcal{A}$ from $\boldsymbol{x}$ if it points inwards $\text{conv}(\mathcal{A})$, (i.e. $\boldsymbol{r} \in \text{cone}(\mathcal{A} - \boldsymbol{x})$).
We define the *pyramidal width* of a set $\mathcal{A}$ to be the smallest pyramidal width of all its faces, i.e.

$$PWidth(\mathcal{A}) := \min_{\substack{\mathcal{K} \in \text{faces}(\text{conv}(\mathcal{A})) \\ \boldsymbol{x} \in \mathcal{K} \\ \boldsymbol{r} \in \text{cone}(\mathcal{K} - \boldsymbol{x}) \setminus \{\mathbf{0}\}}} PdirW(\mathcal{K} \cap \mathcal{A}, \boldsymbol{r}, \boldsymbol{x}). \quad (9)$$

**Theorem 3.** *Let $\boldsymbol{x} \in \mathcal{M} = \text{conv}(\mathcal{A})$ be a suboptimal point and $\mathcal{S}$ be an active set for $\boldsymbol{x}$. Let $\boldsymbol{x}^*$ be an optimal point and corresponding error direction $\hat{\boldsymbol{e}} = (\boldsymbol{x}^* - \boldsymbol{x}) / \|\boldsymbol{x}^* - \boldsymbol{x}\|$, and negative gradient $\boldsymbol{r} := -\nabla f(\boldsymbol{x})$ (and so $\langle \boldsymbol{r}, \hat{\boldsymbol{e}} \rangle > 0$). Let $\boldsymbol{d} = \boldsymbol{s} - \boldsymbol{v}$ be the pairwise FW direction obtained over $\mathcal{A}$ and $\mathcal{S}$ with negative gradient $\boldsymbol{r}$. Then*

$$\frac{\langle \boldsymbol{r}, \boldsymbol{d} \rangle}{\langle \boldsymbol{r}, \hat{\boldsymbol{e}} \rangle} \geq PWidth(\mathcal{A}). \quad (10)$$

## 3.1 Properties of Pyramidal Width and Consequences

**Examples of Values.** The pyramidal width of a set $\mathcal{A}$ is lower bounded by the minimal width over all subsets of atoms, and thus is strictly greater than zero if the number of atoms is finite. On the other hand, this lower bound is often too loose to be useful, as in particular, vertex subsets of the unit cube in dimension $d$ can have exponentially small width $O(d^{-\frac{d}{2}})$ [see Corollary 27 in 36]. On the other hand, as we show here, the pyramidal width of the unit cube is actually $1/\sqrt{d}$, justifying why we kept the tighter but more involved definition (9). See Appendix B.1 for the proof.

**Lemma 4.** *The pyramidal width of the unit cube in $\mathbb{R}^d$ is $1/\sqrt{d}$.*

For the probability simplex with $d$ vertices, the pyramidal width is actually the same as its width, which is $2/\sqrt{d}$ when $d$ is even, and $2/\sqrt{d - 1/d}$ when $d$ is odd [2] (see Appendix B.1). In contrast, the pyramidal width of an infinite set can be zero. For example, for a curved domain, the set of active atoms $\mathcal{S}$ can contain vertices forming a very narrow pyramid, yielding a zero width in the limit.

**Condition Number of a Set.** The inverse of the rate constant $\rho$ appearing in Theorem 1 is the product of two terms: $L/\mu$ is the standard *condition number* of the objective function appearing in the rates of gradient methods in convex optimization. The second quantity $(M/\delta)^2$ (diameter over pyramidal width) can be interpreted as a *condition number* of the domain $\mathcal{M}$, or its *eccentricity*. The more eccentric the constraint set (large diameter compared to its pyramidal width), the slower the convergence. The best condition number of a function is when its level sets are spherical; the analog in term of the constraint sets is actually the regular simplex, which has the maximum width-to-diameter ratio amongst all simplices [see Corollary 1 in 2]. Its eccentricity is (at most) $d/2$. In contrast, the eccentricity of the unit cube is $d^2$, which is much worse.

We conjecture that the pyramidal width of a set of *vertices* (i.e. extrema of their convex hull) is *non-increasing* when another vertex is added (assuming that all previous points remain vertices). For example, the unit cube can be obtained by iteratively adding vertices to the regular probability simplex, and the pyramidal width thereby decreases from $2/\sqrt{d}$ to $1/\sqrt{d}$. This property could provide lower bounds for the pyramidal width of more complicated polytopes, such as $1/\sqrt{d}$ for the $d$-dimensional marginal polytope, as it can be obtained by removing vertices from the unit cube.

**Complexity Lower Bounds.** Combining the convergence Theorem 1 and the condition number of the unit simplex, we get a complexity of $O(d\frac{L}{\mu}\log(\frac{1}{\epsilon}))$ to reach $\epsilon$-accuracy when optimizing a strongly convex function over the unit simplex. Here the linear dependence on $d$ should not come as a surprise, in view of the known lower bound of $1/t$ for $t \leq d$ for Frank-Wolfe type methods [15].

**Applications to Submodular Minimization.** See Appendix A.2 for a consequence of our linear rate for the popular MNP algorithm for submodular function optimization (over the base polytope).

## 4 Non-Strongly Convex Generalization

Building on the work of Beck and Shtern [4] and Wang and Lin [33], we can generalize our global linear convergence results for all Frank-Wolfe variants for the more general case where $f(\boldsymbol{x}) := g(\boldsymbol{A}\boldsymbol{x}) + \langle \boldsymbol{b}, \boldsymbol{x} \rangle$, for $\boldsymbol{A} \in \mathbb{R}^{p \times d}$, $\boldsymbol{b} \in \mathbb{R}^d$ and where $g$ is $\mu_g$-strongly convex and continuously differentiable over $\boldsymbol{A}\mathcal{M}$. We note that for a general matrix $\boldsymbol{A}$, $f$ is convex but not necessarily *strongly* convex. In this case, the linear convergence still holds but with the constant $\mu$ appearing in the rate of Theorem 1 replaced with the generalized constant $\tilde{\mu}$ appearing in Lemma 9 in Appendix F.

## 5 Illustrative Experiments

We illustrate the performance of the presented algorithm variants in two numerical experiments, shown in Figure 2. The first example is a constrained Lasso problem ($\ell_1$-regularized least squares regression), that is $\min_{\boldsymbol{x} \in \mathcal{M}} f(\boldsymbol{x}) = \|\boldsymbol{A}\boldsymbol{x} - \boldsymbol{b}\|^2$, with $\mathcal{M} = 20 \cdot L_1$ a scaled $L_1$-ball. We used a random Gaussian matrix $\boldsymbol{A} \in \mathbb{R}^{200 \times 500}$, and a noisy measurement $\boldsymbol{b} = \boldsymbol{A}\boldsymbol{x}^*$ with $\boldsymbol{x}^*$ being a sparse vector with 50 entries $\pm 1$, and 10% of additive noise. For the $L_1$-ball, the linear minimization oracle $\text{LMO}_{\mathcal{A}}$ just selects the column of $\boldsymbol{A}$ of best inner product with the residual vector. The second application comes from video co-localization. The approach used by [16] is formulated as a quadratic program (QP) over a flow polytope, the convex hull of paths in a network. In this application, the linear minimization oracle is equivalent to finding a shortest path in the network, which can be done easily by dynamic programming. For the $\text{LMO}_{\mathcal{A}}$, we re-use the code provided by [16] and their included aeroplane dataset resulting in a QP over 660 variables. In both experiments, we see that the modified FW variants (away-steps and pairwise) outperform the original FW algorithm, and exhibit a linear convergence. In addition, the constant in the convergence rate of Theorem 1 can also be empirically shown to be fairly tight for AFW and PFW by running them on an increasingly obtuse triangle (see Appendix E).

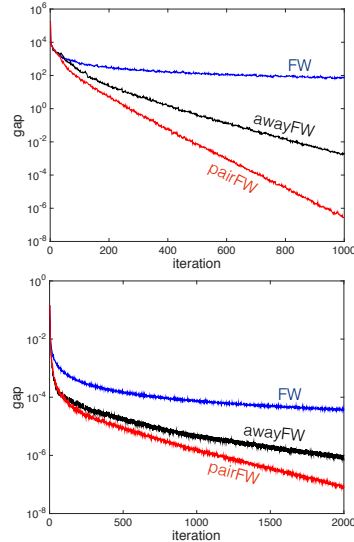

Figure 2: Duality gap $g_t^{\text{FW}}$ vs iterations on the Lasso problem (top), and video co-localization (bottom). Code is available from the authors' website.

**Discussion.** Building on a preliminary version of our work [20], Beck and Shtern [4] also proved a linear rate for away-steps FW, but with a simpler lower bound for the LHS of (10) using linear duality arguments. However, their lower bound [see e.g. Lemma 3.1 in 4] is looser: they get a $d^2$ constant for the eccentricity of the regular simplex instead of the tighter $d$ that we proved. Finally, the recently proposed generic scheme for *accelerating* first-order optimization methods in the sense of Nesterov from [24] applies directly to the FW variants given their global linear convergence rate that we proved. This gives for the first time first-order methods that *only use linear oracles* and obtain the "near-optimal" $\tilde{O}(1/k^2)$ rate for smooth convex functions, or the accelerated $\tilde{O}(\sqrt{L/\mu})$ constant in the linear rate for strongly convex functions. Given that the constants also depend on the dimensionality, it remains an open question whether this acceleration is practically useful.

**Acknowledgements.** We thank J.B. Alayrac, E. Hazan, A. Hubard, A. Osokin and P. Marcotte for helpful discussions. This work was partially supported by the MSR-Inria Joint Center and a Google Research Award.

## Footnotes

[1]The atoms *do not* have to be extreme points (vertices) of $\mathcal{M}$.

[2]All our convergence results can be carefully extended to approximate linear minimization oracles with multiplicative approximation guarantees; we state them for exact oracles in this paper for simplicity.

[3]The original algorithm presented in [34] was not convergent; this was corrected by Guélat and Marcotte [12], assuming a tractable representation of $\mathcal{M}$ with linear inequalities and called it the modified Frank-Wolfe (MFW) algorithm. Our description in Algorithm 1 extends it to the more general setup of (1).

[4]For AFW and PFW, we actually require that $\nabla f$ is $L$-Lipschitz over the larger domain $\mathcal{M} + \mathcal{M} - \mathcal{M}$.

[5]By *proper* convex combination, we mean that all coefficients are non-zero in the convex combination.

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
