[Supplementary Material]

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

(\boldsymbol{x}^{(t)}) - f(\boldsymbol{x}^*) \geq \Omega\big(1/t^{1+\delta}\big)$ for any $\delta > 0$ when $\boldsymbol{x}^*$ lies on a face of $\mathcal{M}$ with some additional regularity assumptions. Note that this lower bound is different than the $\Omega\big(1/t\big)$ one presented in [15, Lemma 3] which holds for all one-atom-per-step algorithms but assumes high dimensionality $d \geq t$.

# 1   Improved Variants of the Frank-Wolfe Algorithm

---

**Algorithm 1** Away-steps Frank-Wolfe algorithm: $\mathbf{AFW}(\boldsymbol{x}^{(0)}, \mathcal{A}, \epsilon)$

---

1: Let $\boldsymbol{x}^{(0)} \in \mathcal{A}$, and $\mathcal{S}^{(0)} := \{\boldsymbol{x}^{(0)}\}$      *(so that $\alpha_{\boldsymbol{v}}^{(0)} = 1$ for $\boldsymbol{v} = \boldsymbol{x}^{(0)}$ and $0$ otherwise)*
2: **for** $t = 0 \ldots T$ **do**
3:     Let $\boldsymbol{s}_t := \mathrm{LMO}_{\mathcal{A}}\big(\nabla f(\boldsymbol{x}^{(t)})\big)$ and $\boldsymbol{d}_t^{\mathrm{FW}} := \boldsymbol{s}_t - \boldsymbol{x}^{(t)}$      *(the FW direction)*
4:     Let $\boldsymbol{v}_t \in \arg\max_{\boldsymbol{v} \in \mathcal{S}^{(t)}} \langle \nabla f(\boldsymbol{x}^{(t)}), \boldsymbol{v} \rangle$ and $\boldsymbol{d}_t^{\mathrm{A}} := \boldsymbol{x}^{(t)} - \boldsymbol{v}_t$      *(the away direction)*
5:     **if** $g_t^{\mathrm{FW}} := \big\langle -\nabla f(\boldsymbol{x}^{(t)}), \boldsymbol{d}_t^{\mathrm{FW}} \big\rangle \leq \epsilon$ **then return** $\boldsymbol{x}^{(t)}$      *(FW gap is small enough, so return)*
6:     **if** $\big\langle -\nabla f(\boldsymbol{x}^{(t)}), \boldsymbol{d}_t^{\mathrm{FW}} \big\rangle \geq \big\langle -\nabla f(\boldsymbol{x}^{(t)}), \boldsymbol{d}_t^{\mathrm{A}} \big\rangle$ **then**
7:         $\boldsymbol{d}_t := \boldsymbol{d}_t^{\mathrm{FW}}$, and $\gamma_{\max} := 1$      *(choose the FW direction)*
8:     **else**
9:         $\boldsymbol{d}_t := \boldsymbol{d}_t^{\mathrm{A}}$, and $\gamma_{\max} := \alpha_{\boldsymbol{v}_t}/(1 - \alpha_{\boldsymbol{v}_t})$      *(choose away direction; maximum feasible step-size)*
10:     **end if**
11:     Line-search: $\gamma_t \in \arg\min_{\gamma \in [0, \gamma_{\max}]} f\big(\boldsymbol{x}^{(t)} + \gamma \boldsymbol{d}_t\big)$
12:     Update $\boldsymbol{x}^{(t+1)} := \boldsymbol{x}^{(t)} + \gamma_t \boldsymbol{d}_t$      *(and accordingly for the weights $\boldsymbol{\alpha}^{(t+1)}$, see text)*
13:     Update $\mathcal{S}^{(t+1)} := \{\boldsymbol{v} \in \mathcal{A} \text{ s.t. } \alpha_{\boldsymbol{v}}^{(t+1)} > 0\}$
14: **end for**

---

**Algorithm 2** Pairwise Frank-Wolfe algorithm: $\mathbf{PFW}(\boldsymbol{x}^{(0)}, \mathcal{A}, \epsilon)$

---

1: ... as in Algorithm 1, except replacing lines 6 to 10 by:  $\boldsymbol{d}_t = \boldsymbol{d}_t^{\mathrm{PFW}} := \boldsymbol{s}_t - \boldsymbol{v}_t$, and $\gamma_{\max} := \alpha_{\boldsymbol{

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

[6]Chakrabarty et al. [39] quoted a complexity of $O(d^7 F^2)$ for MNP. However, this fell short of the earlier result of Bach [3] for classic FW in the submodular minimization case, which was better by two $O(d)$ factors. [39] counted $O(d^3)$ per iteration of the MNP algorithm whereas Wolfe had provided a $O(d^2)$ implementation; and they missed that there were at least $t/2$ good cycles ('non drop steps') after $t$ iterations, rather than $O(t/d)$ as they have used.

[7]This is assuming that the eccentricity of the base polytope does not depend on $F$, which remains to be proven.

[8]We thank Jean-Baptiste Alayrac for inspiring us to use symmetry in the proof.

[9]As a reminder, we define a *k-face* of $\mathcal{M}$ (a $k$-dimensional face of $\mathcal{M}$) a set $\mathcal{K}$ such that $\mathcal{K} = \mathcal{M} \cap \{y : \langle r, y - x \rangle = 0\}$ for some normal vector $r$ and fixed reference point $x \in \mathcal{K}$ with the additional property that $\mathcal{M}$ lies on one side of the given half-space determined by $r$ i.e. $\langle r, y - x \rangle \leq 0 \; \forall y \in \mathcal{M}$. $k$ is the dimensionality of the affine hull of $\mathcal{K}$. We call a $k$-face of dimensions $k = 0, 1, \dim(\mathcal{M}) - 2$ and $\dim(\mathcal{M}) - 1$ a *vertex*, *edge*, *ridge* and *facet* respectively. $\mathcal{M}$ is a $k$-face of itself with $k = \dim(\mathcal{M})$. See Definition 2.1 in the book of Ziegler [41], which we also recommend for more background material on polytopes.

[10]As we are working with general polytopes, the expansion of a point as a convex combination of atoms is not necessarily unique.

[11]As an example of function that is not strongly convex but can still have $\mu_f^A > 0$, consider $f(\boldsymbol{x}) := g(\boldsymbol{A}\boldsymbol{x})$ where $g$ is $\mu_g$-strongly convex, but the matrix $\boldsymbol{A}$ is rank deficient. Then by using the affine invariance of the definition of $\mu_f^A$ and using Theorem (6) applied on the equivalent problem on $g$ with domain $\mathrm{conv}(\boldsymbol{A}\mathcal{A})$, we get $\mu_f^A \geq \mu_g \cdot (PWidth(\boldsymbol{A}\mathcal{A}))^2 > 0$.

[12]Note that any step with $\gamma_{\max} \geq 1$ can also be considered a 'good step', even if $\gamma_t = \gamma_{\max}$, as is apparent from the proof. The problematic steps arise only when $\gamma_{\max} \ll 1$.

[13]Here we have used the trivial inequality $0 \le a^2 - 2ab + b^2$ for the choice of numbers $a := \frac{g_t}{\mu_f^{\mathrm{A}}}$ and $b := \overline{\gamma}$.

[14]Moreover, as we do not have the factor of 2 relating $\langle r_t, d_t^{FW} \rangle$ and $g_t$ unlike in the AFW and approximate FCFW case, we can remove the factor of $\frac{1}{2}$ in front of $g_t$ in (31), removing the factor of $\frac{1}{4}$ appearing in (32), and also giving a geometric decrease with factor $\left(1 - \frac{1}{2}\right)$ when $\gamma_t^B > 1$.

[15] $\boldsymbol{x}_0$ is obtained by taking a random convex combination of the corners of the domain.

[16]Here we have again used the trivial inequality $0 \leq a^2 - 2ab + b^2$ for the choice of numbers $a := \frac{g_t}{\tilde{\mu}_f}$ and $b := \hat{\gamma}$.

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

# Appendix

## A   More on Frank-Wolfe Algorithm Variants

### A.1   Wolfe's Min-Norm Point (MNP) algorithm

A generalization of Wolfe's min-norm point (MNP) algorithm [35] for general convex functions is to run Algorithm 3 with the correction subroutine in step 7 implemented as presented below in Algorithm 5. In Wolfe's paper [35], the correction step is called the minor cycle; whereas the FW outer loop is called the major cycle.

As we have mentioned in Section 1, MNP for polytope distance is often confused with fully-corrective FW as presented in Algorithm 3, for quadratic objectives. In fact, standard FCFW optimizes $f$ over $\mathrm{conv}(\mathcal{A}^{(t)})$, whereas MNP implements the correction as a sequence of *affine* projections on the active set that potentially yield a different update.

---

**Algorithm 5** Generalized version of Wolfe's MNP correction: **MNP-Correction**$(\boldsymbol{x}^{(t)}, \mathcal{A}^{(t)}, \boldsymbol{s}_t)$

---

1: Let $\mathcal{S}^{(0)} := \mathcal{A}^{(t)} \bigcup \{\boldsymbol{s}_t\}$, and $\boldsymbol{z}_0 := \boldsymbol{x}^{(t)}$. Note that $\boldsymbol{x}^{(t)} = \sum_{\boldsymbol{v} \in \mathcal{A}^{(t)}} \alpha_{\boldsymbol{v}}\, \boldsymbol{v}$ and we assume that the elements of $\mathcal{A}^{(t)}$ are *affinely independent*.
2: **for** $k = 1 \ldots |\mathcal{S}^{(0)}|$ **do**
3:     Let $\boldsymbol{y}_k$ be the minimizer of $f$ on the affine hull of $\mathcal{S}^{(k-1)}$
4:     **if** $\boldsymbol{y}_k$ is in the relative interior of $\mathrm{conv}(\mathcal{S}^{(k-1)})$ **then**
5:         **return** $(\boldsymbol{y}_k, \mathcal{S}^{(k-1)})$      ($\mathcal{S}^{(k-1)}$ *is active set for* $\boldsymbol{y}_k$)
6:     **else**
7:         Let $\boldsymbol{z}_k$ be the solution of doing line-search from $\boldsymbol{z}_{k-1}$ to $\boldsymbol{y}_k$. [step 2(d)(iii) of Alg. 1 in 39]
8:         *(Note that $\boldsymbol{z}_k$ now lies on the boundary of $\mathrm{conv}(\mathcal{S}^{(k-1)})$, and so some atoms were removed)*
9:         Let $\mathcal{S}^{(k)}$ be the (affinely independent) active atoms in the expansion of $\boldsymbol{z}_k$.
10:    **end if**
11: **end for**

---

There are two main differences between FCFW and the MNP algorithm. First, after a correction step, MNP guarantees that $\boldsymbol{x}^{(t+1)}$ is *both* the minimizer of $f$ over the *affine hull* of $\mathcal{A}^{(t+1)}$ and also $\mathrm{conv}(\mathcal{A}^{(t+1)})$ (where $\mathcal{A}^{(t+1)}$ might be much smaller than $\mathcal{A}^{(t)} \cup \{\boldsymbol{s}_t\}$), whereas FCFW guarantees that $\boldsymbol{x}^{(t+1)}$ is the minimizer of $f$ over $\mathrm{conv}(\mathcal{A}^{(t)} \cup \{\boldsymbol{s}_t\})$ – this is usually not the case for MNP unless at most one atom was dropped from the correction polytope, as is apparent from our convergence proof. Secondly, the correction atoms $\mathcal{A}^{(t)}$ are always affinely independent for MNP and are identical to the active set $\mathcal{S}^{(t)}$, whereas FCFW can use both redundant as well as inactive atoms. The advantage of the MNP implementation using affine hull projections is that the correction can be efficiently implemented when $f$ is the Euclidean norm, especially when a triangular array repre-

sentation of the active set is maintained (see the careful implementation details in Wolfe's original paper [35]).

The MNP variant indeed only makes sense when the minimization of $f$ over the affine hull of $\mathcal{M}$ is well-defined (and is efficient). Note though that the line-search in step 7 does not require any new information about $\mathcal{M}$, as it is made only with respect to $\mathrm{conv}(\mathcal{S}^{(k-1)})$, for which we have an explicit list of vertices. This line-search can be efficiently computed in $O(|\mathcal{S}^{(k-1)}|)$, and is well described for example in step 2(d)(iii) of Algorithm 1 of Chakrabarty et al. [39].

### A.2  Applications to Submodular Minimization

An interesting consequence of our global linear convergence result for FW algorithm variants here is the potential to reduce the gap between the known theoretical rates and the impressive empirical performance of MNP for submodular function minimization (over the base polytope). While Bach [3] already showed convergence of FW in this case, Chakrabarty et al. [39] later gave a weaker convergence rate for Wolfe's MNP variant. For exact submodular function optimization, the overall complexity by [39] was $O(d^5 F^2)$ (with some corrections[6]), where $F$ is the maximum absolute value of the integer-valued submodular function. This is in contrast to $O(d^5 \log(d\,F))$ for the fastest algorithms [40]. Using our linear convergence, the $F$ factor can be put back in the $\log$ term for MNP,[7] matching their empirical observations that the MNP algorithm was not too sensitive to $F$. The same follows for AFW and FCFW, which is novel.

### A.3  Pairwise Frank-Wolfe

Our new analysis of the pairwise Frank-Wolfe variant as introduced in Section 1 is motivated by the work of Garber and Hazan [10], who provided the first variant of Frank-Wolfe with a global linear convergence rate with explicit constants that do not depend on the location of the optimum $\boldsymbol{x}^*$, for a more complex extension of such a pairwise algorithm. An important contribution of the work of Garber and Hazan [10] was to define the concept of *local linear oracle*, which (approximately) minimizes a linear function on the intersection of $\mathcal{M}$ and a small ball around $\boldsymbol{x}^{(t)}$ (hence the name *local*). They showed that if such a local linear oracle was available, then one could replace the step that moves towards $\boldsymbol{s}_t$ in the standard FW procedure with a constant step-size move towards the point returned by the local linear oracle to obtain a globally linearly convergent algorithm. They then demonstrated how to implement such a local linear oracle by using only one call to the linear oracle (to get $\boldsymbol{s}_t$), as well as sorting the atoms in $\mathcal{S}^{(t)}$ in decreasing order of their inner product with $\nabla f(\boldsymbol{x}^{(t)})$ (note that the first element then is the away atom $\boldsymbol{v}_t$ from Algorithm 1). The procedure implementing the local linear oracle amounts to iteratively swapping the mass from the away atom $\boldsymbol{v}_t$ to the FW atom $\boldsymbol{s}_t$ until enough mass has been moved (given by some precomputed constants). If the amount of mass to move is bigger than $\alpha_{\boldsymbol{v}_t}^{(t)}$, then one sets $\alpha_{\boldsymbol{v}_t}^{(t)}$ to zero and start moving mass from the *second* away atom, and so on, until enough mass has been moved (which is why the sorting is needed). We call such a swap of mass between the away atom and the FW atom a *pairwise FW* step, i.e. $\alpha_{\boldsymbol{v}_t}^{(t+1)} = \alpha_{\boldsymbol{v}_t}^{(t)} - \gamma$ and $\alpha_{\boldsymbol{s}_t}^{(t+1)} = \alpha_{\boldsymbol{s}_t}^{(t)} + \gamma$ for some step-size $\gamma \leq \gamma_{\max} := \alpha_{\boldsymbol{v}_t}^{(t)}$. The local linear oracle is implemented as a sequence of pairwise FW steps, always keeping the same FW atom $\boldsymbol{s}_t$ as the target, but updating the away atom to move from as we set their coordinates to zero.

A major disadvantage of the algorithm presented by Garber and Hazan [10] is that their algorithm is *not adaptive*: it requires the computation of several (loose) constants to determine the step-sizes, which means that the behavior of the algorithm is stuck in its worst-case analysis. The pairwise Frank-Wolfe variant is obtained by simply doing one line-search in the pairwise Frank-Wolfe direction $\boldsymbol{d}_t^{\mathrm{PFW}} := \boldsymbol{s}_t - \boldsymbol{v}_t$ (see Algorithm 2). This gives a fully adaptive algorithm, and it turns out that this is sufficient to yield a global linear convergent rate.

**Notes on Convergence Proofs in [26].** We here point out some corrections to the convergence proofs given in [26] for a variant of pairwise FW that chooses between a standard FW step and a pairwise FW step by picking the one which makes the most progress on the objective after a line-search. [26, Proposition 1] states the global convergence of their algorithm by arguing that $\langle -\nabla f(\boldsymbol{x}^{(t)}), \boldsymbol{d}_t^{\text{PFW}} \rangle \geq \langle -\nabla f(\boldsymbol{x}^{(t)}), \boldsymbol{d}_t^{\text{FW}} \rangle$ and then stating that they can re-use the same pattern as the standard FW convergence proof but with the direction $\boldsymbol{d}_t^{\text{PFW}}$. But this is forgetting the fact that the maximal step-size $\gamma_{\max} = \alpha_{\boldsymbol{v}_t}$ for a pairwise FW step can be too small to make sufficient progress. Their global convergence statement is still correct as every step of their algorithm makes more progress than a FW step, which already has a global convergence result, but this is not the argument they made. Similarly, they state a global linear convergence result in their Proposition 4, citing a proof from [37]. On the other hand, the relevant used Proposition 3 in [37] forgot to consider the possibility of problematic *swap steps* that we had to painfully bound in our convergence Theorem 8; they only considered drop steps or 'good steps', thereby missing a bound on the number of swap steps to get a valid global bound.

### A.4 Other Related Work

Very recently, following the earlier workshop version of our article [20], Pena et al. [28] presented an alternative geometric quantity measuring the linear convergence speed of the AFW algorithm variant. Their approach is motivated by a special case of the Frank-Wolfe method, the von Neumann algorithm. Their complexity constant – called the restricted width – is also bounded away from zero, but its value does depend on the location of the optimal solution, which is a disadvantage shared with the earlier existing results of [34, 12, 5], as well as the line of work of [1, 19, 26] that relies on Robinson's condition [30]. More precisely, the bound on the constant given in [28, Theorem 4] applies to the translated atoms $\tilde{A}$ relative to the optimum point. The constant is not affine-invariant, whereas the constants $\mu_f^A$ (22) and $C_f^A$ (26) in our setting are so, see the discussion in Section C. It would still be interesting to compare the value of our respective constants on standard polytopes.

## B Pyramidal Width

### B.1 Pyramidal Width of the Cube and Probability Simplex

**Lemma' 4.** *The pyramidal width of the unit cube in $\mathbb{R}^d$ is $1/\sqrt{d}$.*

*Proof of Lemma 4.* First consider a point $\boldsymbol{x}$ in the interior of the cube, and let $\boldsymbol{r}$ be the unit length direction achieving the smallest pyramidal width for $\boldsymbol{x}$. Let $\boldsymbol{s} = \boldsymbol{s}(A, \boldsymbol{r})$ (the FW atom in direction $\boldsymbol{r}$).[8] Without loss of generality, by symmetry, we can rotate the cube so that $\boldsymbol{s}$ lies at the origin. This implies that each coordinate of $\boldsymbol{r}$ is non-positive. Represent a vertex $\boldsymbol{v}$ of the cube as its set of indices for which $v_i = 1$. Then $\langle \boldsymbol{r}, \boldsymbol{s} - \boldsymbol{v} \rangle = \sum_{i \in \boldsymbol{v}} -r_i \geq \max_{i \in \boldsymbol{v}} |r_i|$. Consider any possible active set $\mathcal{S}$; as $\boldsymbol{x}$ has all its coordinate strictly positive, for each dimension $i$, there must exist an element of $\mathcal{S}$ with its $i$ coordinate equals to 1. This means that $\max_{\boldsymbol{v} \in \mathcal{S}} \langle \boldsymbol{r}, \boldsymbol{s} - \boldsymbol{v} \rangle \geq \|\boldsymbol{r}\|_\infty$. But as $\boldsymbol{r}$ has unit Euclidean norm, then $\|\boldsymbol{r}\|_\infty \geq 1/\sqrt{d}$. Now consider $\boldsymbol{x}$ to lie on a facet of the cube (i.e. the active set $\mathcal{S}$ is lower dimensional); and let $I := \{i : r_i < 0\}$. Since $\boldsymbol{r}$ has to be feasible from $\boldsymbol{x}$, for each $i \in I$, we cannot have $x_i = 0$ and thus there exists an element of the active set with its $i^{\text{th}}$ coordinate equal to 1. We thus have that $\max_{\boldsymbol{v} \in \mathcal{S}} \langle \boldsymbol{r}, \boldsymbol{s} - \boldsymbol{v} \rangle \geq \|\boldsymbol{r}\|_\infty \geq 1/\sqrt{|I|} \geq 1/\sqrt{d}$. Using the same argument on a lower dimensional $\mathcal{K}$ give a lower bound of $1/\sqrt{\dim(\mathcal{K})}$ which is bigger. These cover all the possibilities appearing in the definition of the pyramidal width, and thus the lower bound is correct. It is achieved by choosing an $\boldsymbol{x}$ in the interior, the canonical basis as the active set $\mathcal{S}$, and the direction defined by $r_i = -1/\sqrt{d}$ for each $i$. $\qquad\square$

We note that both the active set definition $\mathcal{S}$ and the feasibility condition on $\boldsymbol{r}$ were crucially used in the above proof to obtain such a large value for the pyramidal width of the unit cube, thus justifying the somewhat involved definition appearing in (9). On the other hand, the astute reader might have noticed that the important quantity to lower bound for the linear convergence rate of the different FW variants is $\frac{\langle \boldsymbol{r}_t, \boldsymbol{d}_t \rangle}{\langle \boldsymbol{r}_t, \hat{\boldsymbol{e}}_t \rangle}$ (as in (5)), rather than the looser value $\frac{1}{M} \frac{\langle \boldsymbol{r}_t, \boldsymbol{d}_t \rangle}{\langle \boldsymbol{r}_t, \hat{\boldsymbol{e}}_t \rangle}$ that we used to handle the

proof of the difficult Theorem 3 (where we recall that $M$ is the diameter of $\mathcal{M}$). One could thus hope to get a tighter measure for the condition number of a set by considering $\|s - v\|$ (with $s$ and $v$ the minimizing witnesses for the pyramidal width) instead of the diameter $M$ in the ratio diameter / pyramidal width. This might give a tighter constant for general sets, but in the case of the cube, it does not change the general $\Omega(d^2)$ dependence for its condition number. To see this, suppose that $d$ is even and let $k = d/2$. Consider the direction $r$ with $r_i := -1$ for $1 \le i \le k$, and $r_i := -\epsilon$ for $(k+1) \le i \le d$. We thus have that the FW atom $s(\mathcal{A}, r)$ is the origin as before. Consider $x$ such that $x_i := 1/k$ for $1 \le i \le k$, and $x_i := 1$ for $(k+1) \le i \le d$, that is, $x$ is the uniform convex combination of the $k$ vertices which has only one non-zero in the first $k$ coordinates, and the last $k$ coordinates all equal to 1. We have that $r$ is a feasible direction from $x$, and that all vertices in the active set for $x$ have the same inner product with $r$: $\max_{v \in \mathcal{S}}\langle r, s - v \rangle = 1 + k\epsilon$. We thus have:

$$\left\langle \frac{r}{\|r\|}, \frac{s - v}{\|s - v\|} \right\rangle = \frac{1 + k\epsilon}{(\sqrt{k}\sqrt{1 + \epsilon^2})(\sqrt{k + 1})} \le \frac{1}{k} \quad \text{for } \epsilon \text{ small enough.}$$

Squaring the inverse, we thus get that the condition number of the cube is at least $k^2 = d^2/4$ even using this tighter definition, thus not changing the $\Omega(d^2)$ dependence.

**Pyramidal Width for the Probability Simplex.** For any $x$ in the relative interior of the probability simplex on $d$ vertices, we have that $\mathcal{S} = \mathcal{A}$, and thus the pyramidal directional width (8) in the feasible direction $r$ with base point $x$ is the same as the standard directional width. Moreover, any face of the probability simplex is just a probability simplex in lower dimensions (with bigger width). This is why the pyramidal width of the probability simplex is the same as its standard width. The width of a regular simplex was implicitly given in [2]; we provide more details here on this citation. Alexander [2] considers a regular simplex with $k$ vertices and side length $\Delta$. For any partition of the $k$ points into a set of $r$ and $k - r$ points (for $r \le \lfloor k/2 \rfloor$), one can compute the distance $c(r, \Delta)$ between the flats (affine hulls) of the two sets (which also corresponds to a specific directional width). Alexander gives a formula on the third line of p. 91 in [2] for the square of this distance:

$$(c(r, \Delta))^2 = \Delta^2 \frac{k}{2r(k - r)}. \tag{11}$$

The width of the regular simplex is obtained by taking the minimum of (11) with respect to $r \le \lfloor k/2 \rfloor$. As (11) is a decreasing function up to $r = k/2$, we obtain its minimum by substituting $r = \lfloor k/2 \rfloor$. By using $\Delta = \sqrt{2}$, $k = d$ and $r = \lfloor d/2 \rfloor$ in (11), we get that the width for the probability simplex on $d$ vertices is $2/\sqrt{d}$ when $d$ is even and the slightly bigger $2/\sqrt{d - 1/d}$ when $d$ is odd.

From the pyramidal width perspective, one can obtain these numbers by considering any relative interior point of the probability simplex as $x$ and considering the following feasible $r$. For $d$ even, we let $r_i := 1$ for $1 \le i \le d/2$ and $r_i := -1$ for $i > d/2$. Note that $\sum_i r_i = 0$ and thus $r$ is a feasible direction for a point in the relative interior of the probability simplex. Then $\max_{s,v \in \mathcal{A}}\langle \frac{r}{\|r\|}, s - v \rangle = \frac{1}{\sqrt{d}}(1 + 1) = \frac{2}{\sqrt{d}}$ as claimed. For $d$ odd, we can choose $r_i := \frac{2}{d-1}$ for $1 \le i \le \frac{d-1}{2}$, and $r_i := \frac{2}{d+1}$ for $i \ge \frac{d+1}{2}$. Then $\|r\| = \sqrt{\frac{4d}{d^2-1}}$ and $\max_{s,v \in \mathcal{A}}\langle \frac{r}{\|r\|}, s - v \rangle = \sqrt{\frac{4d}{d^2-1}} = 2/\sqrt{d - 1/d}$ as claimed. Showing that these obtained values were the minimum possible ones is non-trivial though, which is why we appealed to the width of the regular simplex computed from [2].

## B.2 Proof of Theorem 3 on the Pyramidal Width

In this section, we prove the main technical result in our paper: a geometric lower bound for the crucial quantity appearing in the linear convergence rate for the FW optimization variants.

**Theorem' 3.** *Let $x \in \mathcal{M} = \text{conv}(\mathcal{A})$ be a suboptimal point and $\mathcal{S}$ be an active set for $x$. Let $x^*$ be an optimal point and corresponding error direction $\hat{e} = (x^* - x)/\|x^* - x\|$, and negative gradient $r := -\nabla f(x)$ (and so $\langle r, \hat{e} \rangle > 0$). Let $d^{\text{PFW}}$ be the pairwise FW direction obtained over $\mathcal{A}$ and $\mathcal{S}$ with negative gradient $r$. Then we have:*

$$\frac{\langle r, d^{\text{PFW}} \rangle}{\langle r, \hat{e} \rangle} \ge PWidth(\mathcal{A}). \tag{12}$$

Figure 3: Depiction of the quantities in the proof of Lemma 5. If $r \notin \text{cone}(\mathcal{K})$, but $r \in \text{span}(\mathcal{K})$, then the unit vector direction $\hat{e}^*$ minimizing the angle with $r$ is generated by a point $x^*$ lying on a facet $\mathcal{K}'$ of the polytope $\mathcal{K}$ that contains $x$.

We first give a proof sketch, and then give the full proof.

Recall that a direction $r$ is *feasible* for $\mathcal{A}$ from $x$ if it points inwards $\text{conv}(\mathcal{A})$, i.e. $r \in \text{cone}(\mathcal{A} - x)$.

*Warm-Up Proof Sketch.* By Cauchy-Schwarz, the denominator of (12) is at most $\|r\|$. If $r$ is a feasible direction from $x$ for $\mathcal{A}$, then the LHS is lower bounded by $PdirW(\mathcal{A}, r, x)$ as $d^{\text{PFW}}$ is included as a possible $s - v$ direction considered in the definition of $PdirW$ (8). If $r$ is not feasible from $x$, this means that $x$ lies on the boundary of $\mathcal{M}$. One can then show that the potential $x^*$ that can maximize $\langle r, \hat{e} \rangle$ has also to lie on a facet $\mathcal{K}$ of $\mathcal{M}$ containing $x$ (see Lemma 5 below). The idea is then to project $r$ onto $\mathcal{K}$, and re-do the argument with $\mathcal{K}$ replacing $\mathcal{M}$ and show that the inequality is in the right direction. This explains why all the subfaces of $\mathcal{M}$ are considered in the definition of the pyramidal width (9), and that only *feasible* directions are considered. $\qquad\square$

**Lemma 5** (Minimizing angle is on a facet). *Let $x$ be at the origin, inside a polytope $\mathcal{K}$ and suppose that $r \in span(\mathcal{K})$ is not a feasible direction for $\mathcal{K}$ from $x$ (i.e. $r \notin cone(\mathcal{K})$). Then a feasible direction in $\mathcal{K}$ minimizing the angle with $r$ lies on a facet[9] $\mathcal{K}'$ of $\mathcal{K}$ that includes the origin $x$. That is:*

$$\max_{e \in \mathcal{K}} \left\langle r, \frac{e}{\|e\|} \right\rangle = \max_{e \in \mathcal{K}'} \left\langle r, \frac{e}{\|e\|} \right\rangle = \max_{e \in \mathcal{K}'} \left\langle r', \frac{e}{\|e\|} \right\rangle \tag{13}$$

*where $\mathcal{K}'$ contains $x$, $\|\cdot\|$ is the Euclidean norm and $r'$ is defined as the orthogonal projection of $r$ on $span(\mathcal{K}')$.*

*Proof.* This seems like an obvious geometric fact (see Figure 3), but we prove it formally, as sometimes high dimensional geometry is tricky (for example, the result is false without the assumption that $r \in \text{span}(\mathcal{K})$ or if $\|\cdot\|$ is not the Euclidean norm). Rewrite the optimization variable on the LHS of (13) as $\hat{e} = \frac{e}{\|e\|}$. The optimization domain for $\hat{e}$ is thus the intersection between the unit sphere and $\text{cone}(\mathcal{K})$. We now show that any maximizer $\hat{e}^*$ cannot lie in the relative interior of $\text{cone}(\mathcal{K})$, and thus it has to lie on a *facet* of $\text{cone}(\mathcal{K})$, implying then that a corresponding maximizer $e^*$ is lying on a facet of $\mathcal{K}$ containing $x$, concluding the proof for the first equality in (13).

First, as $r \in \text{span}(\mathcal{K})$, we can consider without loss of generality that $\text{cone}(\mathcal{K})$ is full dimensional by projecting on its affine hull if needed. We want to solve $\max_{\hat{e}} \langle r, \hat{e} \rangle$ s.t. $\|\hat{e}\|^2 = 1$ and $\hat{e} \in \text{cone}(\mathcal{K})$.

By contradiction, we suppose that $\hat{e}^*$ lies in the interior of $\mathrm{cone}(\mathcal{K})$, and so we can remove the polyhedral cone constraint. The gradient of the objective is the constant $r$ and the gradient of the equality constraint is $2\hat{e}$. By the Karush-Kuhn-Tucker (KKT) necessary conditions for a stationary point to the problem with the only equality constraint $\|\hat{e}\|^2 = 1$ (see e.g. [Proposition 3.3.1 in 38]), then the gradient of the objective is collinear to the gradient of the equality constraint, i.e. we have $\hat{e}^* = \pm\hat{r}$. Since $\hat{r}$ is not feasible, then $\hat{e}^* = -\hat{r}$, which is actually a local *minimum* of the inner product by Cauchy-Schwarz. We thus conclude that the maximizing $\hat{e}^*$ lies on the boundary of $\mathrm{cone}(\mathcal{K})$, concluding the proof for the first equality in (13).

For the second equality in (13), we simply use the fact that $r - r'$ is orthogonal to the elements of $\mathcal{K}'$ by the definition of the orthogonal projection. $\qquad\square$

*Proof of Theorem 3.* Let $\hat{e}(x^*) := \frac{x^* - x}{\|x^* - x\|}$ be the normalized error vector. We consider the worst-case possibility for $x^*$. As $x$ is not optimal, we require that $\langle r, \hat{e}(x^*)\rangle > 0$. We recall that by definition of the pairwise FW direction:

$$\left\langle \frac{r}{\|r\|}, d^{\mathrm{PFW}} \right\rangle = \max_{s \in \mathcal{A}, v \in \mathcal{S}} \left\langle \frac{r}{\|r\|}, s - v \right\rangle \geq \min_{\mathcal{S}' \in \mathcal{S}_x} \max_{s \in \mathcal{A}, v \in \mathcal{S}'} \left\langle \frac{r}{\|r\|}, s - v \right\rangle = PdirW(\mathcal{A}, r, x).$$
(14)

By Cauchy-Schwarz, we always have $\langle r, \hat{e}(x^*)\rangle \leq \|r\|$. If $r$ is a feasible direction from $x$ in $\mathrm{conv}(\mathcal{A})$, then $r$ appears in the set of directions considered in the definition of the pyramidal width (9) for $\mathcal{A}$ and so from (14), we have that the inequality (12) holds.

If $r$ is not a feasible direction, then we iteratively project it on the faces of $\mathcal{M}$ until we get a feasible direction $r'$, obtaining a term $PdirW(\mathcal{A} \cap \mathcal{K}, r', x)$ for some face $\mathcal{K}$ of $\mathcal{M}$ as appearing in the definition of the pyramidal width (9). The rest of the proof formalizes this process. As $x$ is fixed, we work on the centered polytope at $x$ to simplify the statements, i.e. let $\tilde{\mathcal{M}} := \mathcal{M} - x$. We have the following worst case lower bound for (12):

$$\frac{\langle r, d^{\mathrm{PFW}}\rangle}{\langle r, \hat{e}\rangle} \geq \left( \max_{s \in \mathcal{A}, v \in \mathcal{S}} \langle r, s - v\rangle \right) \left( \max_{e \in \tilde{\mathcal{M}}} \left\langle r, \frac{e}{\|e\|} \right\rangle \right)^{-1}.$$
(15)

The first term on the RHS of (15) just comes from the definition of $d^{\mathrm{PFW}}$ (with equality), whereas the second term is considering the worst case possibility for $x^*$ to lower bound the LHS. Note also that the second term has to be strictly greater to zero since $x$ is not optimal.

Without loss of generality, we can assume that $r \in \mathrm{span}(\tilde{\mathcal{M}})$ (otherwise, just project it), as any orthogonal component would not change the inner products appearing in (15). If (this projected) $r$ is feasible from $x$, then $\max_{e \in \tilde{\mathcal{M}}} \langle r, \frac{e}{\|e\|}\rangle = \|r\|$, and we again have the lower bound (14) arising in the definition of the pyramidal width.

We thus now suppose that $r$ is not feasible. By the Lemma 5, we have the existence of a facet $\mathcal{K}'$ of $\tilde{\mathcal{M}}$ that includes the origin $x$ such that:

$$\max_{e \in \tilde{\mathcal{M}}} \left\langle r, \frac{e}{\|e\|} \right\rangle = \max_{e \in \mathcal{K}'} \left\langle r, \frac{e}{\|e\|} \right\rangle = \max_{e \in \mathcal{K}'} \left\langle r', \frac{e}{\|e\|} \right\rangle,$$
(16)

where $r'$ is the result of the orthogonal projection of $r$ on $\mathrm{span}(\mathcal{K}')$. We now look at how the numerator of (15) transforms when considering $r'$ and $\mathcal{K}'$:

$$\begin{aligned}
\max_{s \in \mathcal{A}, v \in \mathcal{S}} \langle r, s - v\rangle &= \max_{s \in \mathcal{M}} \langle r, s - x\rangle + \max_{v \in \mathcal{S}} \langle -r, v - x\rangle \\
&\geq \max_{s \in (\mathcal{K}' + x)} \langle r, s - x\rangle + \max_{v \in \mathcal{S} \cap (\mathcal{K}' + x)} \langle -r, v - x\rangle \\
&= \max_{s \in (\mathcal{K}' + x)} \langle r', s - x\rangle + \max_{v \in \mathcal{S}} \langle -r', v - x\rangle \\
&= \max_{s \in \mathcal{A} \cap (\mathcal{K}' + x), v \in \mathcal{S}} \langle r', s - v\rangle.
\end{aligned}$$
(17)

To go from the first to the second line, we use the fact that the first term yields an inequality as $(\mathcal{K}' + x) \subseteq (\tilde{\mathcal{M}} + x) = \mathcal{M}$. Also, since $x$ is in the relative interior of $\mathrm{conv}(\mathcal{S})$ (as $x$ is a *proper* convex combination of elements of $\mathcal{S}$ by definition), we have that $(\mathcal{S} - x) \subseteq \mathcal{K}$ for any face $\mathcal{K}$ of $\tilde{\mathcal{M}}$

containing the origin $\boldsymbol{x}$. Thus $\mathcal{S} = \mathcal{S} \cap (\mathcal{K}' + \boldsymbol{x})$, and the second term on the first line actually yields an equality for the second line. The third line uses the fact that $\boldsymbol{r} - \boldsymbol{r}'$ is orthogonal to members of $\mathcal{K}'$, as $\boldsymbol{r}'$ is obtained by orthogonal projection.

Plugging (16) and (17) into the inequality (15), we get:

$$\frac{\langle \boldsymbol{r}, \boldsymbol{d}^{\text{PFW}} \rangle}{\langle \boldsymbol{r}, \hat{\boldsymbol{e}} \rangle} \geq \left( \max_{\substack{\boldsymbol{s} \in \mathcal{A} \cap (\mathcal{K}' + \boldsymbol{x}), \\ \boldsymbol{v} \in \mathcal{S}}} \langle \boldsymbol{r}', \boldsymbol{s} - \boldsymbol{v} \rangle \right) \left( \max_{\boldsymbol{e} \in \mathcal{K}'} \langle \boldsymbol{r}', \frac{\boldsymbol{e}}{\|\boldsymbol{e}\|} \rangle \right)^{-1}. \tag{18}$$

We are back to a similar situation to (15), with the lower dimensional $\mathcal{K}'$ playing the role of the polytope $\tilde{\mathcal{M}}$, and $\boldsymbol{r}' \in \text{span}(\mathcal{K}')$ playing the role of $\boldsymbol{r}$. If $\boldsymbol{r}'$ is feasible from $\boldsymbol{x}$ in $\mathcal{K}'$, then re-using the previous argument, we get $PdirW(\mathcal{A} \cap (\mathcal{K}' + \boldsymbol{x}), \boldsymbol{r}', \boldsymbol{x})$ as the lower bound, which is part of the definition of the pyramidal width of $\mathcal{A}$ (note that we have $(\mathcal{K}' + \boldsymbol{x})$ as $\mathcal{K}'$ is a face of the *centered* polytope $\tilde{\mathcal{M}}$). Otherwise (if $\boldsymbol{r} \notin \text{cone}(\mathcal{K}')$), then we use Lemma 5 again to get a facet $\mathcal{K}''$ of $\mathcal{K}'$ as well as a new direction $\boldsymbol{r}''$ which is the orthogonal projection of $\boldsymbol{r}'$ on $\text{span}(\mathcal{K}'')$ such that we can re-do the manipulations for (16) and (18), yielding $PdirW(\mathcal{A} \cap (\mathcal{K}'' + \boldsymbol{x}), \boldsymbol{r}'', \boldsymbol{x})$ as a lower bound if $\boldsymbol{r}''$ is feasible from $\boldsymbol{x}$ in $\mathcal{K}''$. As long as we do not obtain a feasible direction, we keep re-using Lemma 5 to project the direction on a lower dimensional face of $\tilde{\mathcal{M}}$ that contains $\boldsymbol{x}$. This process must stop at some point; ultimately, we will reach the lowest dimensional face $\mathcal{K}_{\boldsymbol{x}}$ that contains $\boldsymbol{x}$. As $\boldsymbol{x}$ lies in the relative interior of $\mathcal{K}_{\boldsymbol{x}}$, then all directions in $\text{span}(\mathcal{K}_{\boldsymbol{x}})$ are feasible, and so the projected $\boldsymbol{r}$ will have to be feasible. Moreover, by stringing together the equalities of the type (16) for all the projected directions, we know that $\max_{\boldsymbol{e} \in \mathcal{K}_{\boldsymbol{x}}} \langle \boldsymbol{r}_{\text{final}}, \frac{\boldsymbol{e}}{\|\boldsymbol{e}\|} \rangle > 0$ (as we originally had $\langle \boldsymbol{r}, \hat{\boldsymbol{e}} \rangle > 0$), and thus $\mathcal{K}_{\boldsymbol{x}}$ is at least one-dimensional and we also have $\boldsymbol{r}_{\text{final}} \neq \boldsymbol{0}$ (this last condition is crucial to avoid having a lower bound of zero!). This concludes the proof, and also explains why in the definition of the pyramidal width (9), we consider the pyramidal directional width for all the faces of $\text{conv}(\mathcal{A})$ and respective non-zero feasible direction $\boldsymbol{r}$. $\qquad\square$

## C  Affine Invariant Formulation

Here we provide linear convergence proofs in terms of affine invariant quantities, since all the Frank-Wolfe algorithm variants presented in this paper are affine invariant. The statements presented in the main paper above are special cases of the following more general theorems, by using the bounds (20) for the curvature constant $C_f$, and Theorem 6 for the affine invariant strong convexity $\mu_f^{\text{A}}$.

An optimization method is called *affine invariant* if it is invariant under affine transformations of the input problem: If one chooses any re-parameterization of the domain $\mathcal{M}$ by a *surjective* linear or affine map $\boldsymbol{A} : \hat{\mathcal{M}} \to \mathcal{M}$, then the "old" and "new" optimization problems $\min_{\boldsymbol{x} \in \mathcal{M}} f(\boldsymbol{x})$ and $\min_{\hat{\boldsymbol{x}} \in \hat{\mathcal{M}}} \hat{f}(\hat{\boldsymbol{x}})$ for $\hat{f}(\hat{\boldsymbol{x}}) := f(\boldsymbol{A}\hat{\boldsymbol{x}})$ look completely the same to the algorithm.

More precisely, every "new" iterate must remain exactly the transform of the corresponding old iterate; an affine invariant analysis should thus yield the convergence rate and constants unchanged by the transformation. It is well known that Newton's method is affine invariant under invertible $\boldsymbol{A}$, and the Frank-Wolfe algorithm and all the variants presented here are affine invariant in the even stronger sense under arbitrary surjective $\boldsymbol{A}$ [15]. (This is directly implied if the algorithm and all constants appearing in the analysis only depend on inner products with the gradient, which are preserved since $\nabla \hat{f} = \boldsymbol{A}^T \nabla f$.)

Note however that the property of being an extremum point (vertex) of $\mathcal{M}$ is *not* affine invariant (see [4, Section 3.1] for an example). This explains why we presented all algorithms here as working with atoms $\mathcal{A}$ rather than vertices of the domain, thus maintaining the affine invariance of the algorithms as well as their convergence analysis.

**Affine Invariant Measures of Smoothness.**  The affine invariant convergence analysis of the standard Frank-Wolfe algorithm by [15] crucially relies on the following measure of non-linearity of the objective function $f$ over the domain $\mathcal{M}$. The (upper) *curvature constant* $C_f$ of a convex and differentiable function $f : \mathbb{R}^d \to \mathbb{R}$, with respect to a compact domain $\mathcal{M}$ is defined as

$$C_f := \sup_{\substack{\boldsymbol{x}, \boldsymbol{s} \in \mathcal{M}, \, \gamma \in [0,1], \\ \boldsymbol{y} = \boldsymbol{x} + \gamma(\boldsymbol{s} - \boldsymbol{x})}} \frac{2}{\gamma^2} \left( f(\boldsymbol{y}) - f(\boldsymbol{x}) - \langle \nabla f(\boldsymbol{x}), \boldsymbol{y} - \boldsymbol{x} \rangle \right). \tag{19}$$

The definition of $C_f$ closely mimics the fundamental descent lemma (2). The assumption of bounded curvature $C_f$ closely corresponds to a Lipschitz assumption on the gradient of $f$. More precisely, if $\nabla f$ is $L$-Lipschitz continuous on $\mathcal{M}$ with respect to some arbitrary chosen norm $\|.\|$ in dual pairing, i.e. $\|\nabla f(\boldsymbol{x}) - \nabla f(\boldsymbol{y})\|_* \leq L \|\boldsymbol{x} - \boldsymbol{y}\|$, then

$$C_f \leq L \operatorname{diam}_{\|.\|}(\mathcal{M})^2 \,, \tag{20}$$

where $\operatorname{diam}_{\|.\|}(.)$ denotes the $\|.\|$-diameter, see [15, Lemma 7]. While the early papers [9, 8] on the Frank-Wolfe algorithm relied on such Lipschitz constants with respect to a norm, the curvature constant $C_f$ here is affine invariant, does not depend on any norm, and gives tighter convergence rates. The quantity $C_f$ combines the complexity of the domain $\mathcal{M}$ and the curvature of the objective function $f$ into a single quantity. The advantage of this combination is well illustrated in [21, Lemma A.1], where Frank-Wolfe was used to optimize a quadratic function over product of probability simplices with an exponential number of dimensions. In this case, the Lipschitz constant could be exponentially worse than the curvature constant which does take the simplex geometry of $\mathcal{M}$ into account.

**An Affine Invariant Notion of Strong Convexity which Depends on the Geometry of $\mathcal{M}$.** We now present the affine invariant analog of the strong convexity bound (4), which could be interpreted as the *lower* curvature $\mu_f^{\mathrm{A}}$ analog of $C_f$. The role of $\gamma$ in the definition (19) of $C_f$ was to define an affine invariant scale by only looking at proportions over lines (as line segments between $\boldsymbol{x}$ and $\boldsymbol{s}$ in this case). The trick here is to use anchor points in $\mathcal{A}$ in order to define standard lengths (by looking at proportions on lines). These anchor points ($\boldsymbol{s}_f(\boldsymbol{x})$ and $\boldsymbol{v}_f(\boldsymbol{x})$ defined below) are motivated directly from the FW atom and the away atom appearing in the away-steps FW algorithm. Specifically, let $\boldsymbol{x}^*$ be a potential optimal point and $\boldsymbol{x}$ a non-optimal point; thus we have $\langle -\nabla f(\boldsymbol{x}), \boldsymbol{x}^* - \boldsymbol{x} \rangle > 0$ (i.e. $\boldsymbol{x}^* - \boldsymbol{x}$ is a strict descent direction from $\boldsymbol{x}$ for $f$). We then define the positive step-size quantity:

$$\gamma^{\mathrm{A}}(\boldsymbol{x}, \boldsymbol{x}^*) := \frac{\langle -\nabla f(\boldsymbol{x}), \boldsymbol{x}^* - \boldsymbol{x} \rangle}{\langle -\nabla f(\boldsymbol{x}), \boldsymbol{s}_f(\boldsymbol{x}) - \boldsymbol{v}_f(\boldsymbol{x}) \rangle} \,. \tag{21}$$

This quantity is motivated from both (6) and the linear rate inequality (5), and enables to transfer lengths from the error $\boldsymbol{e}_t = \boldsymbol{x}^* - \boldsymbol{x}_t$ to the pairwise FW direction $\boldsymbol{d}_t^{\mathrm{PFW}} = \boldsymbol{s}_f(\boldsymbol{x}_t) - \boldsymbol{v}_f(\boldsymbol{x}_t)$. More precisely, $\boldsymbol{s}_f(\boldsymbol{x}) := \arg\min_{\boldsymbol{v} \in \mathcal{A}} \langle \nabla f(\boldsymbol{x}), \boldsymbol{v} \rangle$ is the standard FW atom. To define the away-atom, we consider all possible expansions of $\boldsymbol{x}$ as a convex combination of atoms.[10] We recall that set of possible active sets is $\mathcal{S}_{\boldsymbol{x}} := \{\mathcal{S} \,|\, \mathcal{S} \subseteq \mathcal{A} \text{ such that } \boldsymbol{x} \text{ is a proper convex combination of all the elements in } \mathcal{S}\}$. For a given set $\mathcal{S}$, we write $\boldsymbol{v}_{\mathcal{S}}(\boldsymbol{x}) := \arg\max_{\boldsymbol{v} \in \mathcal{S}} \langle \nabla f(\boldsymbol{x}), \boldsymbol{v} \rangle$ for the away atom in the algorithm supposing that the current set of active atoms is $\mathcal{S}$. Finally, we define $\boldsymbol{v}_f(\boldsymbol{x}) := \arg\min_{\{\boldsymbol{v} = \boldsymbol{v}_{\mathcal{S}}(\boldsymbol{x}) \,|\, \mathcal{S} \in \mathcal{S}_{\boldsymbol{x}}\}} \langle \nabla f(\boldsymbol{x}), \boldsymbol{v} \rangle$ to be the worst-case away atom (that is, the atom which would yield the smallest away descent).

We then define the *geometric strong convexity* constant $\mu_f^{\mathrm{A}}$ which depends *both* on the function $f$ and the domain $\mathcal{M} = \operatorname{conv}(\mathcal{A})$:

$$\mu_f^{\mathrm{A}} := \inf_{\boldsymbol{x} \in \mathcal{M}} \inf_{\substack{\boldsymbol{x}^* \in \mathcal{M} \\ \text{s.t. } \langle \nabla f(\boldsymbol{x}), \boldsymbol{x}^* - \boldsymbol{x} \rangle < 0}} \frac{2}{\gamma^{\mathrm{A}}(\boldsymbol{x}, \boldsymbol{x}^*)^2} \left( f(\boldsymbol{x}^*) - f(\boldsymbol{x}) - \langle \nabla f(\boldsymbol{x}), \boldsymbol{x}^* - \boldsymbol{x} \rangle \right) \,. \tag{22}$$

## C.1  Lower Bound for the Geometric Strong Convexity Constant $\mu_f^{\mathrm{A}}$

The geometric strong convexity constant $\mu_f^{\mathrm{A}}$, as defined in (22), is affine invariant, since it only depends on the inner products of feasible points with the gradient. Also, it combines both the complexity of the function $f$ and the geometry of the domain $\mathcal{M}$. Theorem 6 allows us to lower bound the constant $\mu_f^{\mathrm{A}}$ in terms of the strong convexity of the objective function, combined with a purely geometric complexity measure of the domain $\mathcal{M}$ (its pyramidal width $PWidth(\mathcal{A})$ (9)). In the following Section D below, we will show the linear convergence of the four variants of the FW algorithm presented in this paper under the assumption that $\mu_f^{\mathrm{A}} > 0$.

In view of the following Theorem 6, we have that the condition $\mu_f^A > 0$ is slightly weaker than the strong convexity of the objective function[11] over a polytope domain (it is implied by strong convexity).

**Theorem 6.** *Let $f$ be a convex differentiable function and suppose that $f$ is $\mu$-strongly convex w.r.t. to the Euclidean norm $\|\cdot\|$ over the domain $\mathcal{M} = \mathrm{conv}(\mathcal{A})$ with strong-convexity constant $\mu \geq 0$. Then*

$$\mu_f^A \geq \mu \cdot (PWidth(\mathcal{A}))^2 \ . \tag{23}$$

*Proof.* By definition of strong convexity with respect to a norm, we have that for any $\boldsymbol{x}, \boldsymbol{y} \in \mathcal{M}$,

$$f(\boldsymbol{y}) - f(\boldsymbol{x}) - \langle \nabla f(\boldsymbol{x}), \boldsymbol{y} - \boldsymbol{x} \rangle \geq \tfrac{\mu}{2} \|\boldsymbol{y} - \boldsymbol{x}\|^2 \ . \tag{24}$$

Using the strong convexity bound (24) with $\boldsymbol{y} := \boldsymbol{x}^*$ on the right hand side of equation (22) (and using the shorthand $\boldsymbol{r_x} := -\nabla f(\boldsymbol{x})$ ), we thus get:

$$\mu_f^A \geq \inf_{\substack{\boldsymbol{x}, \boldsymbol{x}^* \in \mathcal{M} \\ \text{s.t. } \langle \boldsymbol{r_x}, \boldsymbol{x}^* - \boldsymbol{x} \rangle > 0}} \mu \left( \frac{\langle \boldsymbol{r_x}, \boldsymbol{s}_f(\boldsymbol{x}) - \boldsymbol{v}_f(\boldsymbol{x}) \rangle}{\langle \boldsymbol{r_x}, \boldsymbol{x}^* - \boldsymbol{x} \rangle} \|\boldsymbol{x}^* - \boldsymbol{x}\| \right)^2$$

$$= \mu \inf_{\substack{\boldsymbol{x} \neq \boldsymbol{x}^* \in \mathcal{M} \\ \text{s.t. } \langle \boldsymbol{r_x}, \hat{\boldsymbol{e}} \rangle > 0}} \left( \frac{\langle \boldsymbol{r_x}, \boldsymbol{s}_f(\boldsymbol{x}) - \boldsymbol{v}_f(\boldsymbol{x}) \rangle}{\langle \boldsymbol{r_x}, \hat{\boldsymbol{e}}(\boldsymbol{x}^*, \boldsymbol{x}) \rangle} \right)^2, \tag{25}$$

where $\hat{\boldsymbol{e}}(\boldsymbol{x}^*, \boldsymbol{x}) := \frac{\boldsymbol{x}^* - \boldsymbol{x}}{\|\boldsymbol{x}^* - \boldsymbol{x}\|}$ is the unit length feasible direction from $\boldsymbol{x}$ to $\boldsymbol{x}^*$. We are thus taking an infimum over all possible feasible directions starting from $\boldsymbol{x}$ (i.e. which moves within $\mathcal{M}$) with the additional constraint that it makes a positive inner product with the negative gradient $\boldsymbol{r_x}$, i.e. it is a strict descent direction. This is only possible if $\boldsymbol{x}$ is not already optimal, i.e. $\boldsymbol{x} \in \mathcal{M} \setminus \mathcal{X}^*$ where $\mathcal{X}^* := \{\boldsymbol{x}^* \in \mathcal{M} : \langle \boldsymbol{r}_{\boldsymbol{x}^*}, \boldsymbol{x} - \boldsymbol{x}^* \rangle \leq 0 \ \forall \boldsymbol{x} \in \mathcal{M}\}$ is the set of optimal points.

We note that $\boldsymbol{s}_f(\boldsymbol{x}) - \boldsymbol{v}_f(\boldsymbol{x})$ is a valid pairwise FW direction for a specific active set $\mathcal{S}$ for $\boldsymbol{x}$, and so we can re-use (12) from Theorem 3 for the right hand side of (25) to conclude the proof. $\square$

We now proceed to present the main linear convergence result in the next section, using only the mentioned affine invariant quantities.

## D   Linear Convergence Proofs

**Curvature Constants.**   Because of the additional possibility of the away step in Algorithm 1, we need to define the following slightly modified additional curvature constant, which will be needed for the linear convergence analysis of the algorithm:

$$C_f^A := \sup_{\substack{\boldsymbol{x}, \boldsymbol{s}, \boldsymbol{v} \in \mathcal{M}, \\ \gamma \in [0,1], \\ \boldsymbol{y} = \boldsymbol{x} + \gamma(\boldsymbol{s} - \boldsymbol{v})}} \frac{2}{\gamma^2} \big( f(\boldsymbol{y}) - f(\boldsymbol{x}) - \gamma \langle \nabla f(\boldsymbol{x}), \boldsymbol{s} - \boldsymbol{v} \rangle \big) \ . \tag{26}$$

By comparing with $C_f$ (19), we see that the modification is that $\boldsymbol{y}$ is defined with any direction $\boldsymbol{s} - \boldsymbol{v}$ instead of a standard FW direction $\boldsymbol{s} - \boldsymbol{x}$. This allows to use the away direction or the pairwise FW direction even though these might yield some $\boldsymbol{y}$'s which are outside of the domain $\mathcal{M}$ when using $\gamma > \gamma_{\max}$ (in fact, $\boldsymbol{y} \in \mathcal{M}^A := \mathcal{M} + (\mathcal{M} - \mathcal{M})$ in the Minkowski sense). On the other hand, by re-using a similar argument as in [15, Lemma 7], we can obtain the same bound (20) for $C_f^A$, with the only difference that the Lipschitz constant $L$ for the gradient function has to be valid on $\mathcal{M}^A$ instead of just $\mathcal{M}$.

**Remark 7.** *For all pairs of functions $f$ and compact domains $\mathcal{M}$, it holds that $\mu_f^A \leq C_f$ (and $C_f \leq C_f^A$).*

*Proof.* Let $\boldsymbol{x}$ be a vertex of $\mathcal{M}$, so that $\mathcal{S} = \{\boldsymbol{x}\}$. Then $\boldsymbol{x} = \boldsymbol{v}_f(\boldsymbol{x})$. Pick $\boldsymbol{x}^* := \boldsymbol{s}_f(\boldsymbol{x})$ and substitute in the definition for $\mu_f^{\mathrm{A}}$ (22). Then $\gamma^{\mathrm{A}}(\boldsymbol{x}, \boldsymbol{x}^*) = 1$ and so we have $\boldsymbol{y} := \boldsymbol{x}^* = \boldsymbol{x} + \gamma(\boldsymbol{x}^* - \boldsymbol{x})$ with $\gamma = 1$ which can also be used in the definition of $C_f$ (19). Thus, we have $\mu_f^{\mathrm{A}} \leq 2\left(f(\boldsymbol{y}) - f(\boldsymbol{x}) - \langle \nabla f(\boldsymbol{x}), \boldsymbol{y} - \boldsymbol{x} \rangle\right) \leq C_f$. $\qquad\square$

We now give the global linear convergence rates for the four variants of the FW algorithm: away-steps FW (AFW Algorithm 1); pairwise FW (PFW Algorithm 2); fully-corrective FW (FCFW Algorithm 3 with approximate correction as per Algorithm 4); and Wolfe's min-norm point algorithm (Algorithm 3 with MNP-correction given in Algorithm 5). For the AFW, MNP and PFW algorithms, we call a *drop step* when the active set shrinks, i.e. $|S^{(t+1)}| < |S^{(t)}|$. For the PFW algorithm, we also have the possibility of a *swap step*, where $\gamma_t = \gamma_{\max}$ but the size of the active set stays constant $|S^{(t+1)}| = |S^{(t)}|$ (i.e. the mass gets fully swapped from the away atom to the FW atom). We note that a nice property of the FCFW variant is that it does not have any drop steps (it executes both FW steps and away steps simultaneously while guaranteeing enough progress at every iteration).

**Theorem 8.** *Suppose that $f$ has smoothness constant $C_f^{\mathrm{A}}$ ($C_f$ for FCFW and MNP), as well as geometric strong convexity constant $\mu_f^{\mathrm{A}}$ as defined in (22). Then the suboptimality $h_t := f(\boldsymbol{x}^{(t)}) - f(\boldsymbol{x}^*)$ of the iterates of all the four variants of the FW algorithm decreases geometrically at each step that is not a drop step nor a swap step (i.e. when $\gamma_t < \gamma_{\max}$, called a 'good step'[12]), that is*

$$h_{t+1} \leq (1 - \rho_f)\, h_t\,,$$

*where:*

$$\rho_f := \frac{\mu_f^{\mathrm{A}}}{4 C_f^{\mathrm{A}}} \quad \textit{for the AFW algorithm,} \qquad \rho_f := \min\left\{\frac{1}{2}, \frac{\mu_f^{\mathrm{A}}}{C_f^{\mathrm{A}}}\right\} \quad \textit{for the PFW algorithm,}$$

$$\rho_f := \frac{\mu_f^{\mathrm{A}}}{4 C_f} \quad \textit{for the FCFW algorithm,} \qquad \rho_f := \min\left\{\frac{1}{2}, \frac{\mu_f^{\mathrm{A}}}{C_f}\right\} \quad \begin{array}{l}\textit{for the MNP algorithm, or}\\ \textit{FCFW with exact correction.}\end{array}$$

*Moreover, the number of drop steps up to iteration $t$ is bounded by $t/2$. This yields the global linear convergence rate of $h_t \leq h_0 \exp(-\frac{1}{2}\rho_f t)$ for the AFW and MNP variants. FCFW does not need the extra $1/2$ factor as it does not have any bad step. Finally, the PFW algorithm has at most $3|\mathcal{A}|!$ swap steps between any two 'good steps'.*

*If $\mu_f^{\mathrm{A}} = 0$ (i.e. the case of general convex objectives), then all the four variants have a $O(1/k(t))$ convergence rate where $k(t)$ is the number of 'good steps' up to iteration $t$. More specifically, we can summarize the suboptimality bounds for the four variants as:*

$$h_t \leq \frac{4C}{k(t) + 4} \quad \textit{for } k(t) \geq 1,$$

*where $C = 2\mu_f^{\mathrm{A}} + h_0$ for AFW; $C = 2C_f + h_0$ for FCFW with approximate correction; $C = C_f/2$ for MNP; and $C = C_f^{\mathrm{A}}/2$ for PFW. The number of good steps is $k(t) = t$ for FCFW; it is $k(t) \geq t/2$ for MNP and AFW; and $k(t) \geq t/(3|\mathcal{A}|! + 1)$ for PFW.*

*Proof.* **Proof for AFW.** The general idea of the proof is to use the definition of the geometric strong convexity constant to upper bound $h_t$, while using the definition of the curvature constant $C_f^{\mathrm{A}}$ to lower bound the decrease in primal suboptimality $h_t - h_{t+1}$ for the 'good steps' of Algorithm 1. Then we upper bound the number of 'bad steps' (the drop steps).

*Upper bounding $h_t$.* In the whole proof, we assume that $\boldsymbol{x}^{(t)}$ is not already optimal, i.e. that $h_t > 0$. If $h_t = 0$, then because line-search is used, we will have $h_{t+1} \leq h_t = 0$ and so the geometric rate of decrease is trivially true in this case. Let $\boldsymbol{x}^*$ be an optimum point (which is not necessarily unique). As $h_t > 0$, we have that $\left\langle -\nabla f(\boldsymbol{x}^{(t)}), \boldsymbol{x}^* - \boldsymbol{x}^{(t)} \right\rangle > 0$. We can thus apply the geometric strong

convexity bound (22) at the current iterate $\boldsymbol{x} := \boldsymbol{x}^{(t)}$ using $\boldsymbol{x}^*$ as an optimum reference point to get (with $\overline{\gamma} := \gamma^{\mathrm{A}}(\boldsymbol{x}^{(t)}, \boldsymbol{x}^*)$ as defined in (21)):

$$
\begin{aligned}
\frac{\overline{\gamma}^2}{2}\mu_f^{\mathrm{A}} &\le f(\boldsymbol{x}^*) - f(\boldsymbol{x}^{(t)}) + \left\langle -\nabla f(\boldsymbol{x}^{(t)}), \boldsymbol{x}^* - \boldsymbol{x}^{(t)} \right\rangle \qquad (27) \\
&= -h_t + \overline{\gamma}\left\langle -\nabla f(\boldsymbol{x}^{(t)}), \boldsymbol{s}_f(\boldsymbol{x}^{(t)}) - \boldsymbol{v}_f(\boldsymbol{x}^{(t)}) \right\rangle \\
&\le -h_t + \overline{\gamma}\left\langle -\nabla f(\boldsymbol{x}^{(t)}), \boldsymbol{s}_t - \boldsymbol{v}_t \right\rangle \\
&= -h_t + \overline{\gamma}g_t \,,
\end{aligned}
$$

where we define $g_t := \left\langle -\nabla f(\boldsymbol{x}^{(t)}), \boldsymbol{s}_t - \boldsymbol{v}_t \right\rangle$ (note that $h_t \le g_t$ and so $g_t$ also gives a primal suboptimality certificate). For the third line, we have used the definition of $\boldsymbol{v}_f(\boldsymbol{x})$ which implies $\left\langle \nabla f(\boldsymbol{x}^{(t)}), \boldsymbol{v}_f(\boldsymbol{x}^{(t)}) \right\rangle \le \left\langle \nabla f(\boldsymbol{x}^{(t)}), \boldsymbol{v}_t \right\rangle$. Therefore $h_t \le -\frac{\overline{\gamma}^2}{2}\mu_f^{\mathrm{A}} + \overline{\gamma}g_t$, which is always upper bounded[13] by

$$
h_t \le \frac{g_t{}^2}{2\mu_f^{\mathrm{A}}}. \qquad (28)
$$

*Lower bounding progress $h_t - h_{t+1}$.* We here use the key aspect in the proof that we had described in the main text with (6). Because of the way the direction $\boldsymbol{d}_t$ is chosen in the AFW Algorithm 1, we have

$$
\left\langle -\nabla f(\boldsymbol{x}^{(t)}), \boldsymbol{d}_t \right\rangle \ge g_t/2, \qquad (29)
$$

and thus $g_t$ characterizes the quality of the direction $\boldsymbol{d}_t$. To see this, note that $2\left\langle \nabla f(\boldsymbol{x}^{(t)}), \boldsymbol{d}_t \right\rangle \le \left\langle \nabla f(\boldsymbol{x}^{(t)}), \boldsymbol{d}_t^{\mathrm{FW}} \right\rangle + \left\langle \nabla f(\boldsymbol{x}^{(t)}), \boldsymbol{d}_t^{\mathrm{A}} \right\rangle = \left\langle \nabla f(\boldsymbol{x}^{(t)}), \boldsymbol{d}_t^{\mathrm{FW}} + \boldsymbol{d}_t^{\mathrm{A}} \right\rangle = -g_t$.

We first consider the case $\gamma_{\max} \ge 1$. Let $\boldsymbol{x}_\gamma := \boldsymbol{x}^{(t)} + \gamma\boldsymbol{d}_t$ be the point obtained by moving with step-size $\gamma$ in direction $\boldsymbol{d}_t$, where $\boldsymbol{d}_t$ is the one chosen by Algorithm 1. By using $\boldsymbol{s} := \boldsymbol{x}^{(t)} + \boldsymbol{d}_t$ (a feasible point as $\gamma_{\max} \ge 1$), $\boldsymbol{x} := \boldsymbol{x}^{(t)}$ and $\boldsymbol{y} := \boldsymbol{x}_\gamma$ in the definition of the curvature constant $C_f$ (19), and solving for $f(\boldsymbol{x}_\gamma)$, we get the affine invariant version of the descent lemma (2):

$$
f(\boldsymbol{x}_\gamma) \le f(\boldsymbol{x}^{(t)}) + \gamma\left\langle \nabla f(\boldsymbol{x}^{(t)}), \boldsymbol{d}_t \right\rangle + \frac{\gamma^2}{2}C_f, \quad \text{valid } \forall \gamma \in [0,1]. \qquad (30)
$$

As $\gamma_t$ is obtained by line-search and that $[0,1] \subseteq [0, \gamma_{\max}]$, we also have that $f(\boldsymbol{x}^{(t+1)}) = f(\boldsymbol{x}_{\gamma_t}) \le f(\boldsymbol{x}_\gamma) \; \forall \gamma \in [0,1]$. Combining these two inequalities, subtracting $f(\boldsymbol{x}^*)$ on both sides, and using $C_f \le C_f^{\mathrm{A}}$ to simplify the possibilities yields $h_{t+1} \le h_t + \gamma\left\langle \nabla f(\boldsymbol{x}^{(t)}), \boldsymbol{d}_t \right\rangle + \frac{\gamma^2}{2}C_f^{\mathrm{A}}$.

Using the crucial gap inequality (29), we get $h_{t+1} \le h_t - \gamma\frac{g_t}{2} + \frac{\gamma^2}{2}C_f^{\mathrm{A}}$, and so:

$$
h_t - h_{t+1} \ge \gamma\frac{g_t}{2} - \frac{\gamma^2}{2}C_f^{\mathrm{A}} \quad \forall \gamma \in [0,1]. \qquad (31)
$$

We can minimize the bound (31) on the right hand side by letting $\gamma = \gamma_t^{\mathrm{B}} := \frac{g_t}{2C_f^{\mathrm{A}}}$. Supposing that $\gamma_t^{\mathrm{B}} \le 1$, we then get $h_t - h_{t+1} \ge \frac{g_t^2}{8C_f^{\mathrm{A}}}$ (we cover the case $\gamma_t^{\mathrm{B}} > 1$ later). By combining this inequality with the one from geometric strong convexity (28), we get

$$
h_t - h_{t+1} \ge \frac{\mu_f^{\mathrm{A}}}{4C_f^{\mathrm{A}}} h_t \qquad (32)
$$

implying that we have a geometric rate of decrease $h_{t+1} \le \left(1 - \frac{\mu_f^{\mathrm{A}}}{4C_f^{\mathrm{A}}}\right)h_t$ (this is a 'good step').

*Boundary cases.* We now consider the case $\gamma_t^{\mathrm{B}} > 1$ (with $\gamma_{\max} \ge 1$ still). The condition $\gamma_t^{\mathrm{B}} > 1$ then translates to $g_t \ge 2C_f^{\mathrm{A}}$, which we can use in (31) with $\gamma = 1$ to get $h_t - h_{t+1} \ge \frac{g_t}{2} - \frac{g_t}{4} = \frac{g_t}{4}$.

An alternative way to obtain the bound is to look at the unconstrained maximum of the RHS which is a concave function of $\overline{\gamma}$ by letting $\overline{\gamma} = g_t/\mu_f^{\mathrm{A}}$, as we did in the main paper to obtain the upper bound on $h_t$ in (5).

Combining this inequality with $h_t \leq g_t$ gives the geometric decrease $h_{t+1} \leq \left(1 - \frac{1}{4}\right) h_t$ (also a 'good step'). $\rho_f^A$ is obtained by considering the worst-case of the constants obtained from $\gamma_t^B > 1$ and $\gamma_t^B \leq 1$. (Note that $\mu_f^A \leq C_f^A$ by Remark 7, and thus $\frac{1}{4} \geq \frac{\mu_f^A}{4C_f^A}$.)

Finally, we are left with the case that $\gamma_{\max} < 1$. This is thus an away step and so $d_t = d_t^A = x^{(t)} - v_t$. Here, we use the away version $C_f^A$: by letting $s := x^{(t)}$, $v = v_t$ and $y := x_\gamma$ in (26), we also get the bound $f(x_\gamma) \leq f(x^{(t)}) + \gamma \left\langle \nabla f(x^{(t)}), d_t \right\rangle + \frac{\gamma^2}{2} C_f^A$, valid $\forall \gamma \in [0, 1]$ (but note here that the points $x_\gamma$ are not feasible for $\gamma > \gamma_{\max}$ – the bound considers some points outside of $\mathcal{M}$). We now have two options: either $\gamma_t = \gamma_{\max}$ (a drop step) or $\gamma_t < \gamma_{\max}$. In the case $\gamma_t < \gamma_{\max}$ (the line-search yields a solution in the interior of $[0, \gamma_{\max}]$), then because $f(x_\gamma)$ is convex in $\gamma$, we know that $\min_{\gamma \in [0,\gamma_{\max}]} f(x_\gamma) = \min_{\gamma \geq 0} f(x_\gamma)$ and thus $\min_{\gamma \in [0,\gamma_{\max}]} f(x_\gamma) = f(x^{(t+1)}) \leq f(x_\gamma)$ $\forall \gamma \in [0, 1]$. We can then re-use the same argument above equation (31) to get the inequality (31), and again considering both the case $\gamma_t^B \leq 1$ (which yields inequality (32)) and the case $\gamma_t^B > 1$ (which yields $\left(1 - \frac{1}{4}\right)$ as the geometric rate constant), we get a 'good step' with $1 - \rho_f$ as the worst-case geometric rate constant.

Finally, we can easily bound the number of drop steps possible up to iteration $t$ with the following argument (the drop steps are the 'bad steps' for which we cannot show good progress). Let $A_t$ be the number of steps that added a vertex in the expansion (only standard FW steps can do this) and let $D_t$ be the number of drop steps. We have that $|\mathcal{S}^{(t)}| = |\mathcal{S}^{(0)}| + A_t - D_t$. Moreover, we have that $A_t + D_t \leq t$. We thus have $1 \leq |\mathcal{S}^{(t)}| \leq |\mathcal{S}^{(0)}| + t - 2D_t$, implying that $D_t \leq \frac{1}{2}(|\mathcal{S}^{(0)}| - 1 + t) = \frac{t}{2}$, as stated in the theorem.

**Proof for FCFW.** In the case of FCFW, we do not need to consider away steps: by the quality of the approximate correction in Algorithm 4 (as specified in Line 4), we know that at the beginning of a new iteration, the away gap $g_t^A \leq \epsilon$. Supposing that the algorithm does not exit at line 6 of Algorithm 3, then $g_t^{FW} > \epsilon$ and thus we have that $2\langle r_t, d_t^{FW} \rangle \geq \langle r_t, d_t^{PFW} \rangle$ using a similar argument as in (6) (i.e. if one would be to run the AFW algorithm at this point, it would take a FW step). Finally, by property of the line 3 of the approximate correction Algorithm 4, the correction is guaranteed to make at least as much progress as a line-search in direction $d_t^{FW}$, and so the lower bound (31) can be used for FCFW as well (but using $C_f$ as the constant instead of $C_f^A$ given that it was a FW step).

**Proof for MNP.** After a correction step in the MNP algorithm, we have that the current iterate is the minimizer over the active set, and thus $g_t^A = 0$. We thus have $\langle r_t, d_t^{FW} \rangle = \langle r_t, d_t^{PFW} \rangle = g_t$, which means that a standard FW step would yield a geometric decrease of error.[14] It thus remains to show that the MNP-correction is making as much progress as a FW line-search. Consider $y_1$ as defined in Algorithm 5. If it belongs to $\mathrm{conv}(\mathcal{V}^{(0)})$, then it has made more progress than a FW line-search as $s_t$ and $x^{(t)}$ belongs to $\mathrm{conv}(\mathcal{V}^{(0)})$.

The next possibility is the crucial step in the proof: suppose that exactly one atom was removed from the correction polytope and that $y_1$ does not belong to $\mathrm{conv}(\mathcal{V}^{(0)})$ (as this was covered in the above case). This means that $y_2$ belongs to *the relative interior* of $\mathrm{conv}(\mathcal{V}^{(1)})$. Because $y_2$ is by definition the affine minimizer of $f$ on $\mathrm{conv}(\mathcal{V}^{(1)})$, the negative gradient $-\nabla f(y_2)$ is pointing away to the polytope $\mathrm{conv}(\mathcal{V}^{(1)})$ (by the optimality condition). But $\mathrm{conv}(\mathcal{V}^{(1)})$ is a *facet* of $\mathrm{conv}(\mathcal{V}^{(0)})$, this means that $-\nabla f(y_2)$ determines a facet of $\mathrm{conv}(\mathcal{V}^{(0)})$ (i.e. $\langle -\nabla f(y_2), y - y_2 \rangle \leq 0$ for all $y \in \mathrm{conv}(\mathcal{V}^{(0)})$). This means that $y_2$ is also the minimizer of $f$ on $\mathrm{conv}(\mathcal{V}^{(0)})$ and thus has made more progress than a FW line-search.

In the case that two atoms are removed from $\mathrm{conv}(\mathcal{V}^{(0)})$, we cannot make this argument anymore (it is possible that $y_3$ makes less progress than a FW line-search); but in this case, the size of the active set is reduced by one (we have a drop step), and thus we can use the same argument as in the AFW algorithm to bound the number of such steps.

**Proof for PFW.** In this case, $\langle \boldsymbol{r}_t, \boldsymbol{d}_t \rangle = \langle \boldsymbol{r}_t, \boldsymbol{d}_t^{\text{PFW}} \rangle$, so we do not even need a factor of 2 to relate the gaps (with the same consequence as in MNP in getting slightly bigger constants). We can re-use the same argument as in the AFW algorithm to get a geometric progress when $\gamma_t < \gamma_{\max}$. When $\gamma_t = \gamma_{\max}$ we can either have a drop step if $\boldsymbol{s}_t$ was already in $\mathcal{S}^{(t)}$, or a swap step if $\boldsymbol{s}_t$ was also added to $\mathcal{S}^{(t)}$ and so $|\mathcal{S}^{(t+1)}| = |\mathcal{S}^{(t)}|$. The number of drop steps can be bounded similarly as in the AFW algorithm. On the other hand, in the worst case, there could be a very large number of swap steps. We provide here a very loose bound, though it would be interesting to use other properties of the objective to prove that this worst case scenario cannot happen.

We thus bound the maximum number of swap steps between two 'good steps' (very loosely). Let $m = |\mathcal{A}|$ be the number of possible atoms, and let $r$ be the size of the current active set $|\mathcal{S}^{(t)}| = r \le m$. When doing a drop step $\gamma_t = \alpha_{\boldsymbol{v}_t}$, there are two possibilities: either we move all the mass from $\boldsymbol{v}_t$ to a new atom $\boldsymbol{s}_t \notin \mathcal{S}^{(t)}$ i.e. $\alpha_{\boldsymbol{v}_t}^{(t+1)} = 0$ and $\alpha_{\boldsymbol{s}_t}^{(t+1)} = \alpha_{\boldsymbol{v}_t}^{(t)}$ (a swap step); or we move all the mass from $\boldsymbol{v}_t$ to an old atom $\boldsymbol{s}_t \in \mathcal{S}^{(t)}$ i.e. $\alpha_{\boldsymbol{s}_t}^{(t+1)} = \alpha_{\boldsymbol{s}_t}^{(t)} + \alpha_{\boldsymbol{v}_t}^{(t)}$ (a 'full drop step'). When doing a swap step, the set of *possible values* for the coordinates $\alpha_{\boldsymbol{v}}$ *do not change*, they are only 'swapped around' amongst the $m$ possible slots. The maximum number of possible consecutive swap steps without revisiting an iterate already seen is thus bounded by the number of ways we can assign $r$ numbers in $m$ slots (supposing the $r$ coordinates were all distinct in the worst case), which is $m!/(m-r)!$. Note that because the algorithm does a line-search in a strict descent direction at each iteration, we always have $f(\boldsymbol{x}^{(t+1)}) < f(\boldsymbol{x}^{(t)})$ unless $\boldsymbol{x}^{(t)}$ is already optimal. This means that the algorithm cannot revisit the same point unless it has converged. When doing a 'full drop step', the set of coordinates changes, but the size of the active set is reduced by one (thus $r$ reduced by one). In the worst case, we will do a maximum number of swap steps, followed by a full drop step, repeated so on all the way until we reach an active set of only one element (in which case there is a maximum number of $m$ swap steps). Starting with an active set of $r$ coordinates, the maximum number of swap steps $B$ without doing any 'good step' (which would also change the set of coordinates), is thus upper bounded by:

$$ B \le \sum_{l=1}^{r} \frac{m!}{(m-l)!} \le m! \sum_{l=0}^{\infty} \frac{1}{l!} = m!\, e \le 3m! \;, $$

as claimed.

**Proof of Sublinear Convergence for General Convex Objectives (i.e. when $\mu_f^{\text{A}} = 0$).** For the good steps of the MNP algorithm and the pairwise FW algorithm, we have the reduction of suboptimality given by (31) without the factor of $\frac{1}{2}$ in front of $g_t \ge h_t$. This is the standard recurrence that appears in the convergence proof of Frank-Wolfe (see for example Equation (4) in [15, proof of Theorem 1]), yielding the usual convergence:

$$ h_t \le \frac{2C}{k(t)+2} \quad \text{for } k(t) \ge 1, \tag{33} $$

where $k(t)$ is the number of good steps up to iteration $t$, and $C = C_f$ for MNP and $C = C_f^{\text{A}}$ for PFW. The number of good steps for MNP is $k(t) \ge t/2$, while for PFW, we have the (useless) lower bound $k(t) \ge t/(3|\mathcal{A}|! + 1)$. For FCFW with exact correction, the rate (33) was already proven in [15] with $k(t) = t$. On the other hand, for FCFW with approximate correction, and for AFW, the factor of $\frac{1}{2}$ in front of the gap $g_t$ in the suboptimality bound (31) somewhat complicates the convergence proof. The recurrence we get for the suboptimality is the same as in Equation (20) of [21, proof of Theorem C.1], with $\nu = \frac{1}{2}$ and $n = 1$, giving the following suboptimality bound:

$$ h_t \le \frac{4C}{k(t)+4} \quad \text{for } k(t) \ge 0, \tag{34} $$

where $C = 2C_f^{\text{A}} + h_0$ for AFW and $C = 2C_f + h_0$ for FCFW with approximate correction. Moreover, the number of good steps is $k(t) \ge t/2$ for AFW, and $k(t) = t$ for FCFW. A weaker (logarithmic) dependence on the initial error $h_0$ can also be obtained by following a tighter analysis (see [21, Theorem C.4] or [18, Lemma D.5 and Theorem D.6]), though we only state the simpler result here. $\qquad\square$

Figure 4: Simple triangle domain to test the empirical tightness of the constant in the convergence rate for AFW and PFW. The width $\delta$ varies with $\theta$. We optimize $f(\boldsymbol{x}) := \frac{1}{2}\|\boldsymbol{x} - \boldsymbol{x}^*\|^2$ over this domain.

(a) Ratio of empirical rate vs. theoretical one

(b) Ratio of empirical rate of PFW vs. AFW

Figure 5: Empirical rate results for the triangle domain of Figure 4. We plot median values over 20 random starting points; the error bars represent the 25% and 75% quantiles. The empirical rate for PFW is closely following the theoretical one.

## E   Empirical Tightness of Linear Rate Constant

We describe here a simple experiment to test how tight the constant in the linear convergence rate of Theorem 8 is. We test both AFW and PFW on the triangle domain with corners at the locations $(-1,0)$, $(0,0)$ and $(\cos(\theta), \sin(\theta))$, for increasingly small $\theta$ (see Figure 4).

The pyramidal width $\delta = \sin(\frac{\theta}{2})$ becomes vanishingly small as $\theta \to 0$; the diameter is $M = 2\cos(\frac{\theta}{2})$. We consider the optimization of the function $f(\boldsymbol{x}) := \frac{1}{2}\|\boldsymbol{x} - \boldsymbol{x}^*\|^2$ with $\boldsymbol{x}^* = (-0.5, 1)$ on one edge of the domain. Note that the condition number of $f$ is $\frac{L}{\mu} = 1$. The bound on the linear convergence rate $\rho$ according to Theorem 8 (using $C_f^{\mathrm{A}} \leq LM^2$ (20) and $\mu_f^{\mathrm{A}} \geq \mu\,\delta^2$ (23)) is $\rho^{\mathrm{PFW}} = \frac{\mu_f^{\mathrm{A}}}{C_f^{\mathrm{A}}} \geq \frac{\mu}{L}\left(\frac{\delta}{M}\right)^2$ for PFW and $\rho^{\mathrm{A}} = \frac{1}{4}\rho^{\mathrm{PFW}}$ for AFW. The theoretical constant here is thus $\rho^{\mathrm{PFW}} = \frac{1}{4}\tan^2(\frac{\theta}{2})$. We consider $\theta$ varying from $\pi/4$ to $1e-3$, and thus theoretical rates varying on a wide range from $0.04$ to $1e-7$. We compare the theoretical rate $\rho$ with the empirically observed one by estimating $\hat{\rho}$ in the relationship $h_t \approx h_0 \exp(-\rho t)$ (using linear regression on the semilogarithmic scale). For each $\theta$, we run both AFW and PFW for 2000 iterations starting from 20 different random starting points[15] in the interior of the triangle domain. We disregard the starting points that yield a drop step (as then the algorithm converges in one iteration; these happen for about 10% of the starting points). Note that as there is no drop step in our setup, we do not need to divide by two the effective rate as is done in Theorem 8 (the number of 'good steps' is $k(t) = t$).

Figure 5 presents the results. In Figure 5(a), we plot the ratio of the estimated rate over the theoretical rate $\frac{\hat{\rho}}{\rho}$ for both PFW and AFW as $\theta$ varies. Note that the ratio is very stable for PFW (around 10), despite the rate changing through six orders of magnitude, demonstrating the empirical tightness of the constant for this domain. The ratio for AFW has more fluctuations, but also stays within a stable range. We can also do a finer analysis than the pyramidal width and consider the finite number possibilities for the worst case angles for $(\langle \hat{r}, d^{\text{PFW}}(r) \rangle)^2$. This gives the tighter constant $\rho^{\text{PFW}} = \sin^2(\frac{\theta}{2})$ for our triangle domain, gaining a factor of about 4, but still not matching yet the empirical observation for PFW.

In Figure 5(b), we compare the empirical rate for PFW vs. the one for AFW. For bigger theoretical rates, PFW appears to converge faster. However, AFW gets a slightly better empirical rate for very small rates (small angles).

## F  Non-Strongly Convex Generalization

Here we will study the generalized setting with objective $f(x) := g(Ax) + \langle b, x \rangle$ where $g$ is $\mu_g$-*strongly convex* w.r.t. the Euclidean norm over the domain $A\mathcal{M}$ with strong convexity constant $\mu_g > 0$.

We first define a few constants: let $G := \max_{x \in \mathcal{M}} \|\nabla g(Ax)\|$ be the maximal norm of the gradient of $g$ over $A\mathcal{M}$; $M$ be the diameter of $\mathcal{M}$ and $M_A$ be the diameter of $A\mathcal{M}$.

Let $\theta$ be the Hoffman constant (see [4, Lemma 2.2]) associated with the matrix $[A; b^\top; B] = \begin{pmatrix} A \\ b^\top \\ B \end{pmatrix}$, where the rows of $B$ are the linear inequality constraints defining the set $\mathcal{M}$.

We present here a generalization of Lemma 2.5 from [4]:

**Lemma 9.** *For any $x \in \mathcal{M}$ and $x^*$ in the solution set $\mathcal{X}^*$:*

$$f(x^*) - f(x) - 2\langle \nabla f(x), x^* - x \rangle \geq 2\tilde{\mu}\, d(x, \mathcal{X}^*)^2, \tag{35}$$

*where $\tilde{\mu} := 1/\left( 2\theta^2 \left( \|b\|M + 3GM_A + \frac{2}{\mu_g}(G^2 + 1) \right) \right)$ is the generalized strong convexity constant for $f$.*

*Proof.* Let $x^*$ be any element of the solution set $\mathcal{X}^*$. By the strong convexity of $g$, we have

$$f(x^*) - f(x) - \langle \nabla f(x), x^* - x \rangle \geq \tfrac{\mu_g}{2} \|Ax^* - Ax\|^2 . \tag{36}$$

Moreover, by the convexity of $f$, we have:

$$-\langle \nabla f(x), x^* - x \rangle \geq f(x) - f(x^*). \tag{37}$$

We now use inequality (2.10) in [4] to get the bound:

$$f(x) - f(x^*) \geq \frac{1}{B_1}(\langle b, x \rangle - \langle b, x^* \rangle)^2, \tag{38}$$

where $B_1 := (\|b\|M + 3GM_A + \frac{2}{\mu_g}G^2)$.

Plugging (38) into (37) and adding to (36), we get:

$$f(x^*) - f(x) - 2\langle \nabla f(x), x^* - x \rangle \geq \frac{1}{B_2}\left( \|Ax^* - Ax\|^2 + (\langle b, x \rangle - \langle b, x^* \rangle)^2 \right)$$

$$\geq \frac{1}{B_2\theta^2}d(x, \mathcal{X}^*)^2,$$

where $B_2 := (\|b\|M + 3GM_A + \frac{2}{\mu_g}(G^2 + 1))$. For the last inequality, we used inequality (2.1) in [4] that made use of the Hoffman's Lemma (see [4, Lemma 2.2]), where $\theta$ is the Hoffman constant associated with the matrix $[A; b^\top; B]$. In this case, $B$ is the matrix with rows containing the linear inequality constraints defining $\mathcal{M}$. $\qquad\square$

We now define the following generalization of the geometric strong convexity constant (22), that we now call $\tilde{\mu}_f$:

$$\tilde{\mu}_f := \inf_{\boldsymbol{x}\in\mathcal{M}} \sup_{\substack{\boldsymbol{x}^*\in\mathcal{X}^* \\ \text{s.t. } \langle\nabla f(\boldsymbol{x}),\boldsymbol{x}^*-\boldsymbol{x}\rangle<0}} \frac{1}{2\gamma^{\mathrm{A}}(\boldsymbol{x},\boldsymbol{x}^*)^2} \left( f(\boldsymbol{x}^*)-f(\boldsymbol{x})-2\langle\nabla f(\boldsymbol{x}),\boldsymbol{x}^*-\boldsymbol{x}\rangle \right). \qquad (39)$$

Notice the new inner *supremum* over the solution set $\mathcal{X}^*$ compared to the original definition (22), the factor of 2 in front of the gradient, and the different overall scaling to have a similar form as in the previous linear convergence theorem. This new quantity $\tilde{\mu}_f$ is still *affine invariant*, but unfortunately now depends on the location of the solution set $\mathcal{X}^*$. We now present the generalization of Theorem 6.

**Theorem 10.** *Let $f(\boldsymbol{x}) := g(\boldsymbol{A}\boldsymbol{x}) + \langle\boldsymbol{b},\boldsymbol{x}\rangle$ where $g$ is $\mu_g$-strongly convex w.r.t. the Euclidean norm over the domain $\boldsymbol{A}\mathcal{M}$ with strong convexity constant $\mu_g > 0$. Let $\tilde{\mu}$ be the corresponding generalized strong convexity constant coming from Lemma 9. Then*

$$\tilde{\mu}_f \geq \tilde{\mu}\cdot (PWidth(\mathcal{M}))^2 \ .$$

*Proof.* Let $\boldsymbol{x}$ be fixed and not optimal; let $\boldsymbol{x}^*$ be its closest point in $\mathcal{X}^*$ i.e. $\|\boldsymbol{x}-\boldsymbol{x}^*\| = d(\boldsymbol{x},\mathcal{X}^*)$. We have that $\langle\nabla f(\boldsymbol{x}),\boldsymbol{x}^*-\boldsymbol{x}\rangle < 0$ as $\boldsymbol{x}$ is not optimal.

We use the generalized strong convexity notion $\tilde{\mu}$ from Lemma 9 for the particular reference point $\boldsymbol{x}^*$ in the third line below to get:

$$\sup_{\substack{\boldsymbol{x}'\in\mathcal{X}^* \\ \text{s.t. } \langle\nabla f(\boldsymbol{x}),\boldsymbol{x}'-\boldsymbol{x}\rangle<0}} \frac{1}{2\gamma^{\mathrm{A}}(\boldsymbol{x},\boldsymbol{x}')^2} \left( f(\boldsymbol{x}')-f(\boldsymbol{x})-2\langle\nabla f(\boldsymbol{x}),\boldsymbol{x}'-\boldsymbol{x}\rangle \right)$$

$$\geq \frac{1}{2\gamma^{\mathrm{A}}(\boldsymbol{x},\boldsymbol{x}^*)^2} \left( f(\boldsymbol{x}^*)-f(\boldsymbol{x})-2\langle\nabla f(\boldsymbol{x}),\boldsymbol{x}^*-\boldsymbol{x}\rangle \right)$$

$$\geq \frac{1}{2\gamma^{\mathrm{A}}(\boldsymbol{x},\boldsymbol{x}^*)^2} 2\tilde{\mu}\, d(\boldsymbol{x},\mathcal{X}^*)^2 = \frac{1}{\gamma^{\mathrm{A}}(\boldsymbol{x},\boldsymbol{x}^*)^2}\tilde{\mu}\|\boldsymbol{x}-\boldsymbol{x}^*\|^2.$$

We can do this for each non-optimal $\boldsymbol{x}$. We thus obtain:

$$\tilde{\mu}_f \geq \inf_{\substack{\boldsymbol{x},\boldsymbol{x}^*\in\mathcal{M} \\ \text{s.t. } \langle\boldsymbol{r}_{\boldsymbol{x}},\boldsymbol{x}^*-\boldsymbol{x}\rangle>0}} \tilde{\mu}\left( \frac{\langle\boldsymbol{r}_{\boldsymbol{x}},\boldsymbol{s}_f(\boldsymbol{x})-\boldsymbol{v}_f(\boldsymbol{x})\rangle}{\langle\boldsymbol{r}_{\boldsymbol{x}},\boldsymbol{x}^*-\boldsymbol{x}\rangle}\|\boldsymbol{x}^*-\boldsymbol{x}\| \right)^2. \qquad (40)$$

And we are back to the same situation as in the proof of our earlier Theorem 6, the only change being that we now have equation (25) holding for the general strong convexity constant $\tilde{\mu}$ instead of its classical analogue $\mu$. $\qquad\square$

Having this tool at hand, the linear convergence of all Frank-Wolfe algorithm variants now holds with the earlier $\mu_f^{\mathrm{A}}$ complexity constant replaced with $\tilde{\mu}_f$. The factor of 2 in the denominator of (39) is to ensure the same scaling.

Again, as we have shown in Theorem 10, we have that our condition $\tilde{\mu}_f > 0$ leading to linear convergence is slightly weaker than generalized strong convexity in the Hoffman sense (it is implied by it).

**Theorem 11.** *Suppose that $f$ has smoothness constant $C_f^{\mathrm{A}}$ ($C_f$ for FCFW and MNP), as well as generalized geometric strong convexity constant $\tilde{\mu}_f$ as defined in (39).*

*Then the suboptimality error $h_t$ of the iterates of all the four variants of the FW algorithm (AFW, FCFW, MNP and PFW) decreases geometrically at each step that is not a drop step nor a swap step (i.e. when $\gamma_t < \gamma_{\max}$), with the same constants as in Theorem 8, except that $\mu_f^{\mathrm{A}}$ is replaced by $\tilde{\mu}_f$.*

*Proof.* The proof closely follows the proof of Theorem 8.

We start from the above generalization (39) of the original geometric strong convexity constant (22), and first replace the inf over $\boldsymbol{x}$ by considering only the choice $\boldsymbol{x} := \boldsymbol{x}^{(t)}$, giving

$$\tilde{\mu}_f \leq \sup_{\substack{\boldsymbol{x}^*\in\mathcal{X}^* \\ \text{s.t. } \langle\nabla f(\boldsymbol{x}^{(t)}),\boldsymbol{x}^*-\boldsymbol{x}^{(t)}\rangle<0}} \frac{1}{2\gamma^{\mathrm{A}}(\boldsymbol{x}^{(t)},\boldsymbol{x}^*)^2} \left( f(\boldsymbol{x}^*)-f(\boldsymbol{x}^{(t)})-2\langle\nabla f(\boldsymbol{x}^{(t)}),\boldsymbol{x}^*-\boldsymbol{x}^{(t)}\rangle \right). \quad (41)$$

From here, we will now mirror our earlier derivation for an upper bound on the suboptimality as a function of the gap $g_t$, as given in (27). As an optimal reference point $\boldsymbol{x}^*$ in (27), we will choose a $\tilde{\boldsymbol{x}}^*$ attaining the supremum in (41), given $\boldsymbol{x}^{(t)}$.

We again employ the 'step-size quantity' $\overline{\gamma} := \gamma^{\mathrm{A}}(\boldsymbol{x}^{(t)}, \tilde{\boldsymbol{x}}^*)$ as defined in (21). Using (41), we have

$$2\overline{\gamma}^2\, \tilde{\mu}_f \leq f(\boldsymbol{x}^*) - f(\boldsymbol{x}^{(t)}) + 2\left\langle -\nabla f(\boldsymbol{x}^{(t)}), \tilde{\boldsymbol{x}}^* - \boldsymbol{x}^{(t)} \right\rangle \qquad (42)$$

$$= -h_t + 2\overline{\gamma}\left\langle -\nabla f(\boldsymbol{x}^{(t)}), \boldsymbol{s}_f(\boldsymbol{x}^{(t)}) - \boldsymbol{v}_f(\boldsymbol{x}^{(t)}) \right\rangle$$

$$\leq -h_t + 2\overline{\gamma}\left\langle -\nabla f(\boldsymbol{x}^{(t)}), \boldsymbol{s}_t - \boldsymbol{v}_t \right\rangle$$

$$= -h_t + 2\overline{\gamma}g_t\ ,$$

Therefore $h_t \leq -\frac{\hat{\gamma}^2}{2}\tilde{\mu}_f + \hat{\gamma} g_t$ when writing $\hat{\gamma} := 2\overline{\gamma}$, which is always upper bounded[16] by

$$h_t \leq \frac{g_t^2}{2\tilde{\mu}_f}. \qquad (43)$$

which is exactly the bound (28) as in the classical case, with the denominator being $2\tilde{\mu}_f$ instead of $2\mu_f^{\mathrm{A}}$.

From here, the proof of the main convergence Theorem 8 continues without modification, using $\tilde{\mu}_f$ instead of $\mu_f^{\mathrm{A}}$.  □

## Supplementary References

[37] H. Allende, E. Frandi, R. Nanculef, and C. Sartori. A novel Frank-Wolfe algorithm. Analysis and applications to large-scale SVM training. *arXiv:1304.1014v2*, 2013.

[38] D. P. Bertsekas. *Nonlinear programming*. Athena Scientific, second edition, 1999.

[39] D. Chakrabarty, P. Jain, and P. Kothari. Provable submodular minimization using Wolfe's algorithm. In *NIPS*. 2014.

[40] S. Iwata. A faster scaling algorithm for minimizing submodular functions. In *Integer Programming and Combinatorial Optimization*, volume 2337 of *Lecture Notes in Computer Science*, pages 1–8. 2002.

[41] G. M. Ziegler. *Lectures on Polytopes*, volume 152 of *Graduate Texts in Mathematics*. Springer, 1995.