[Reviews · NeurIPS 2015]

Submitted by Assigned_Reviewer_1

Summary of paper:

This paper proves linear convergence of (four) variants of the Frank-Wolfe algorithm, under weaker conditions than in the existing literature.

Quality:

The paper is well written and the results are interesting. I think overall it would make a decent addition to NIPS, as it is a meaningful advancement of our understanding of the Frank-Wolfe procedure, which is increasingly important eg. for sub modular problems.

Clarity:

The paper is clearly written overall. I feel the reader of an 8-page nips version would be better served with a "in words" description of the different variants, with the exact algorithmic description boxes being in the appendix. As currently written it is hard to parse what precisely the variants are doing differently, one any way needs to rely on the text.

The intuition section is useful.

Originality:

As per this reviewer, the main original contribution of this paper is to present a more unified view of the various FW variants already proposed, and also a more unified proof of their convergence.

Significance:

Overall, this paper will provide a good reference for variants of the FW algorithm. FW algorithms, like other first-order methods that leverage problem structure, have seen renewed interest in the ML community, so this could be interesting to the NIPS audience.
Summary: This paper proves linear convergence of (four) variants of the Frank-Wolfe algorithm, under weaker conditions than in the existing literature. It gives both a unifying view of existing results, and establishes new convergence guarantees.

Submitted by Assigned_Reviewer_2

The paper has some writing issues. There are a couple of typos, missing punctuation signs (like in page 2, "Section 4As A is finite". The reading is pleasant in general until page 4. The description of the Pairwise Frank-Wolfe method is problematic. Repeated words, undefined concepts (weight), weird sentences... There are some other writing problems along page 4, and in the rest of the document. For instance, there are undefined acronyms (like QP).

As mentioned, the theoretical results are interesting, as well as the interpretations.

The experimental section however, is quite weak. Firstly, it's very hard to see anything in Fig 2, and the lines are indistinguishable in printed version.

Secondly, the (experimental) advantages of the variants of the FW algorithms are known. I expected to see some comments on the corroboration of the theoretical results presented in the paper: convergence rate, constants, etc. Finally, I know that the presence of the sparse coding problem is didactic, as a toy example, but it would be correct to add some state-of-the-art methods for the Lasso problem.

Minor comments:

- be consistent in the bibliography: for instance: ArXiv, arXiv, arXiv.org, for three different papers.

- a section with the conclusions would be appreciated

- the content of Section 4 could be placed somewhere through the text, there's no need for a whole section
Summary: In this paper, the authors address the convergence rate for the Frank-Wolfe algorithm and some of its variants.

The theory is well presented (although the general writing of the paper should be significantly improved), and the results seem interesting and powerful. The last two sections have some problems

Submitted by Assigned_Reviewer_3

Although I acknowledge the main thrust of the paper is theoretical, the experimental part is rather weak. In addition to illustrating the improved performance achieved by the several variants of Frank-Wolfe considered, it would also be very useful to give an idea about when these variants become competitive (in practice) with other methods, specially in problems for which many other methods can be used, such as l1-regularized regression in its unconstrained formulation.

Summary: This is a nice paper, providing theoretical support to a number of improvements to the family of Frank-Wolfe algorithms, some of which had been shown in practice to considerably speed up convergence, but still lacked theoretical guarantees. The paper seems solid and well written, apart from a few minor mistakes that the authors can surely correct by carefully proofreading it.

Submitted by Assigned_Reviewer_4

Report for "On the Global Linear Convergence of Frank-Wolfe Optimization Variants".

## Summary

The authors consider the problem of minimizing a strongly convex or partially strongly convex function over a polyhedron. They describe and analyse different variants of the Frank-Wolfe algorithm. The authors show that the different variants of the algorithm allow to bypass the known fact that the vanilla Frank-Wolfe algorithm is not able to take advantage of strong convexity properties of the objective function. This leads to proof of linear convergence for all the variants presented. The analysis is based on the introduction of a combinatorial positive constant that relates to a notion of condition number for polyhedra. Numerical results on both synthetic and real examples are presented.

## Main comments

The paper present various versions of the original algorithm which were introduced previously in the literature. Linear convergence is a new finding for some of the variants and constitutes an interesting contribution. Furthermore, the pyramidal width introduced in the paper may have interesting applications beyond linear oracle based method. The proof arguments are reasonable up to my understanding and I do not see any major issue related to the content of this work. I detail a few comments that should be taken into account in the next paragraph, some of them may need to be reflected in the text. Additional minor comments are given in the following section.

## Detailed comments

- It may be worth mentioning what happens for these algorithms when the strong convexity assumption is not verified.

- There is a lot of emphasis on the notion of affine invariance. I understand the idea and it is important. But I think it could be clarified. For example, I think that the fact that the linear map is surjective is important. Also I do not think that the statement that A is arbitrary in [17] (line 710) is true as the paragraph dedicated to affine invariance in [17] mentions explicitly the surjectivity.

- In the proof of Theorem 3 in the appendix, I think that the conclusion of line 640 is correct but the arguments are not. First, it could be explicitly mentioned what is meant by KKT. Second KKT only expresses stationarity at the first order and I am not sure that this is enough to conclude that the solution of minimizing a linear function on the intersection of a cone and the unit (euclidean) sphere is on the relative boundary of the cone. I do not know what the authors expicitly meant by "KKT", but for example for the cone $\RR_+ \times \RR$, $d= (-1,0)$ does not belong to the cone and $y = (1,0)$ is in the interior of the cone and satisfies some stationarity condition

which I guess correspond more or less to the KKT condition the authors have in mind.

- The MNP variant actually requires to have a subroutine that minimizes the objective over an affine hull. Since in the model, $f$ is only defined on script M, the setting in which this make sense is slightly different. Furthermore, this has very important practical implications in terms of implementation. This could be emphasized. I guess the line search in step 7 involves the constraints. In this case, this is another significant requirement compared to other variants for which this is explicitly avoided (see line 163).

- The authors claim that everything works the same for the partially strongly convex model just by replacing a constant by another. It is a bit fast since the constants in (21) and (34) look quite different, in particular, there are multiplicative 2 factor that do not occur at the same place. Some hints are welcome.

- The away step variant under the partial strongly convex model was treated in [4]. Is there a difference in the estimated rates? It would be nice to have a discussion, for example based on the cases for which the pyramidal width is known.

- For the simplex, the fact that the pyramidal width is the same as the width deserves more justification. Also the presentation would benefit from a more precise citation of [2] (which result the authors refere to in this paper).

- Up to my understanding, the convergence analysis of [12] does not treat the case "strongly convex objective + strongly convex set". This paper actually proposes the linear convergence as an open problem contrary to what is stated in the introduction.

## Minor comments

- The abstract mentions "weaker condition than strong convexity". I would add "of the objective".

- May be add some reference about the convergence rates of projected gradient method

- Line 99: typo "While the tracking the active set"

- Regarding footnote 3, the lower bound actually holds for specific sequences that never reach the face of interest. It does not hold for every sequence, it depends on the initialization and the problem at hand.

- Line (151) mentions that the active set is small, while after a large number of iterations it is potentially large, unless there is some specific step performed to optimize the vertex representation.

- Line 178: "is to in each step only move", I would add commas

- Line 188: typo "due our"

- Line 210: typo "more computed more efficiently"

- Line 212: "overarching"?

- (7) could actually be summarized as $h_t \geq \min{g_t/2, g_t^2/(LM^2)}

- The authors conjecture a nonincreasingness property for the constant they introduced. Do they have an intuition why this should be the case beyond the fact that this constant is greater for smaller for the cube than the simplex?

- Figure 2 is hardly readable

- On page 12 (appendix), the notation $r$ is used in the main text and in footnote 7 to denote two different thing. Similar comment holds for y.

- The argument following (14) go really fast and non trivial details are missing. It should be written explicitly that (14) is equivalent to the problem mentioned on line 640. Here also, $y$ and $y^*$ are used to denote the decision variable and solution of two different problems.

- On line 689, (14) should be (15).

- Line 700 makes a self reference to "the appendix": it is itself part of the appendix.

- Line 950, I guess "drop step" should be "swap step".

- In (33), $B$ should be $B_1$.
Summary: Linear convergence is a new finding for some of the variants presented and constitutes an interesting contribution. I do not see any major issue related to the content of this work.

Author Feedback
Author rebuttal: We thank all the reviewers for their valuable comments which will improve the final version of this paper. We will carefully implement them in the revision.

== Lasso possibly replaced with experiment on tightness of constant? ==
Both Reviewer 2 and Reviewer 6 asked about the state-of-the-art for Lasso solvers. Indeed the inclusion of this example was mainly for didactic purpose, illustrating a typical improvement between the FW variants. We do not claim that FW-type algorithms are the state-of-the-art Lasso solvers in terms of optimization objective (though they have interesting properties in term of sparsity guarantees and support recovery, but this is outside the scope of this paper). In contrast, the second experiment (a QP for video co-localization) is more illustrative for problems with a large but structured constraint set, where FW indeed becomes state-of-the-art, see also the Examples paragraph on p.2.

If the reviewers prefer, we could move the Lasso case to the supplementary material and instead experimentally illustrate the tightness of the convergence rate constant from Theorem 1. This was not included in the submission due to space constraints. In this simple experiment, we compare the theoretical linear rate rho (from Theorem 1) with the empirically observed hat{rho} for MFW and pFW on a triangle domain with increasingly obtuse angles (and thus smaller and smaller pyramidal widths). The triangle being (-1,0), (0,0) and (cos(theta), sin(theta)) with theta getting close to zero; the objective is 0.5*||x-x*||^2 with x* on the boundary at (-1/2,0). Interestingly, we get that hat{rho} / rho for pFW is quite close to a small constant (~ 10) as rho ranges over multiple orders of magnitude (from 1e-1 to 1e-8 e.g.), illustrating the tightness of the rate in the worst case. The variation over multiple random starting points in the triangle is small (as long as it does not start with a drop step, making the algorithm converge in 2 steps).

Alternatively, we would briefly mention the empirical tightness of the linear rate in the worst case in the main text, and refer to this experiment in the supplementary material. We let the reviewers judge which option would be most valuable, given the scope of this paper.

== Questions from Reviewer 3 ==
We thank the reviewer for this especially in-depth and exceptionally valuable review, and answer their questions below.

- Rate for non-strongly convex functions: We'll clarify that all variants inherit the standard sublinear O(1/t) rate of FW in this case.

- Indeed in L710, we meant 'surjective' for A. We'll clarify.

- Proof Thm 3 in Appendix line 640: The reviewer is correct with their example, (1,0) is the only interior point of the cone satisfying the necessary conditions of stationarity (KKT = Karush-Kuhn-Tucker necessary conditions of optimality for non-linear continuously differentiable optimization); but this point is a *global minimum* (and we wanted to maximize the inner product), thus it can be excluded. The only other points satisfying the necessary KKT conditions need to have some inequality constraints active and thus lie on a face of the cone. We will add details for this (or perhaps make it a lemma?).

- MNP comment (Algorithm 5 in the appendix): We will clarify in the appendix that the MNP variant indeed only makes sense when the minimization of f over the affine hull of M is well-defined (and is efficient). Note though that the line-search in step 7 *does not* require any new information about M, as it is made only with respect to conv(V^(k-1)) [this will be clarified], for which we have an explicit list of vertices. This line-search can be efficiently computed in O(|V^(k-1)|), and is well described for example in step 2(d)(iii) of Algorithm 1 of [6].

- Difference with rates of [4]: one part of the constant in [4] is simpler than the pyramidal width, but unfortunately, their overall rate is suboptimal for simplex-like domains. For example, suppose that we write the complexity of MFW as O(C(d) L/mu log(1/eps)), where C(d) is the eccentricity (L378) of the domain in dimension d for our rate. Their rate gives C(d) = d^2 for the regular simplex, l1-ball and unit cube; whereas we get C(d) = d for the regular simplex and l1-ball; and the same C(d) = d^2 for the unit cube.

- (21) - > (34): we were too fast indeed: (34) should have a 1/4 factor instead of 1/2. The new constants need to be put in a similar derivation as in the lines 851 to 855, to get (21) with the same 1/2 factor. We'll clarify and correct.

- You are correct about [12].

- "Non-increasing property conjecture for pyramidal width": we actually have a proof sketch, but the corner cases are tricky and thus we left it as a conjecture for now. The key property used in the proof is to show that the witness base S in (8) that achieves the pyramidal width has to contain vertices that are *adjacent* (in M) to the summit of the pyramid (the FW corner s in (8)).